# Solar ultraviolet B radiation promotes α-MSH secretion to attenuate the function of ILC2s via the pituitary–lung axis

Yuying Huang [1,12], Lin Zhu[1,12], Shipeng Cheng[1,12], Ranran Dai[2,12], Chunrong Huang[2,12], Yanyan Song[3], Bo Peng[2], Xuezhen Li[1], Jing Wen[1], Yi Gong[4], Yunqian Hu[5], Ling Qian[6], Linyun Zhu[7], Fengying Zhang[8], Li Yu[9], Chunyan Yi[1], Wangpeng Gu[1], Zhiyang Ling [1], Liyan Ma[1], Wei Tang [2]✉, Li Peng [10]✉, Guochao Shi [2]✉, Yaguang Zhang [1,11]✉ & Bing Sun [1]✉

The immunomodulatory effects of ultraviolet B (UVB) radiation in human diseases have been described. Whether type 2 lung inflammation is directly affected by solar ultraviolet (UV) radiation is not fully understood. Here, we show a possible negative correlation between solar UVB radiation and asthmatic inflammation in humans and mice. UVB exposure to the eyes induces hypothalamus-pituitary activation and α-melanocyte-stimulating hormone (α-MSH) accumulation in the serum to suppress allergic airway inflammation by targeting group 2 innate lymphoid cells (ILC2) through the MC5R receptor in mice. The α-MSH/MC5R interaction limits ILC2 function through attenuation of JAK/STAT and NF-κB signaling. Consistently, we observe that the plasma α-MSH concentration is negatively correlated with the number and function of ILC2s in the peripheral blood mononuclear cells (PBMC) of patients with asthma. We provide insights into how solar UVB radiation-driven neuroendocrine α-MSH restricts ILC2-mediated lung inflammation and offer a possible strategy for controlling allergic diseases.

Complex environmental cues have profound effects on human health and diseases. Whether immune disorders such as asthma are sensitive to environmental factors, especially solar ultraviolet (UV) radiation, remains to be investigated. Many studies have described the regulatory effects of UVB (290–315 nm) irradiation on experimental allergic disease in mice[1–5]. However, the mechanisms of UVB radiation-induced immune suppression remain debated.

Asthma is a common chronic pulmonary inflammatory disease that affects more than 300 million people worldwide[6]. Emerging evidence has demonstrated that in addition to CD4[+] Th2 cells, group 2

[1]State Key Laboratory of Cell Biology, Center for Excellence in Molecular Cell Science, Shanghai Institute of Biochemistry and Cell Biology, Chinese Academy of Sciences, University of Chinese Academy of Sciences, Shanghai, China. [2]Department of Pulmonary and Critical Care Medicine, Ruijin Hospital, Shanghai Jiao Tong University School of Medicine, Shanghai, China. [3]Department of Biostatistics, Clinical Research Institute, Shanghai Jiao Tong University School of Medicine, Shanghai, China. [4]Huashan Hospital Affiliated to Fudan University, Shanghai, China. [5]Department of Pulmonary and Critical Care Medicine, Shanghai East Hospital, School of Medicine, Tongji University, Shanghai, China. [6]Department of Pulmonary and Critical Care Medicine, Shanghai Fifth People's Hospital, Fudan University, Shanghai, China. [7]Shanghai Putuo District Central Hospital, Shanghai, China. [8]Shanghai Putuo District People's Hospital, Shanghai, China. [9]Department of Pulmonary and Critical Care Medicine, Tongji Hospital, School of Medicine, Tongji University, Shanghai, China. [10]Shanghai Key Laboratory of Meteorology and Health, Shanghai Meteorological Service, Shanghai, China. [11]Med-X Institute, Center for Immunological and Metabolic Diseases, The First Affiliated Hospital of Xi'an JiaoTong University, Xi'an JiaoTong University, Xi'an, Shaanxi, P. R. China. [12]These authors contributed equally: Yuying Huang, Lin Zhu, Shipeng Cheng, Ranran Dai, Chunrong Huang. ✉e-mail: tina_tangwei@163.com; phyllis_pl@163.com; shiguochao@hotmail.com; zhangyaguang@sibcb.ac.cn; bsun@sibs.ac.cn

innate lymphoid cells (ILC2) are an important and much earlier source of type 2 cytokines, such as IL-5, IL-13, amphiregulin (AREG) and IL-4[7,8]. Upon different triggers, such as protease allergens[9,10], fungi[11], and viruses[12], lung epithelial cells secrete IL-33, IL-25, and thymic stromal lymphopoietin (TSLP)[13–15], which activate ILC2 to induce robust IL-5 and IL-13 production to favor the induction of type 2 immune response. These cytokine cascades drive eosinophil infiltration, airway hyperresponsiveness (AHR), and mucus production[9,16]. In addition to alarmins, cytokines[17], lipid mediators[18,19], neurotransmitters[20–27], hormones[28], and nutrients[29,30] have been shown to regulate ILC2 function. These factors cooperatively make ILC2 highly sensitive in the lung microenvironment. The central nervous system (CNS)-mediated release of circulating mediators, such as endogenous opioids and a diverse range of other hormones, can also affect innate and adaptive immune cells[31]. As the "conductor of the endocrine orchestra", the hypothalamic-pituitary (HP) unit has been reported to produce hormones such as prolactin (PRL), adrenocorticotropic hormone (ACTH), and thyroid stimulating hormone (TSH) to regulate immune responses[32]. Moreover, Some environmental factors, such as UV radiation, can activate the HP unit and release downstream hormones[33], suggesting that environmental stress and the neuroendocrine system might modulate the immune responses. Although considerable advances have been made in defining different factors that regulate ILC2 responses, the detailed mechanisms underlying the regulation of ILC2 and type 2 inflammation by environment-driven neuroendocrine in airway inflammation remain defined.

Here, we analyze asthma emergency department (ED) visit distribution data for 17,818 individuals from 2015 to 2019 in Shanghai, China. We find that asthma attacks are significantly associated with solar UVB radiation and its seasonal fluctuation, resulting in a lower incidence of asthma during the summer. Similar results are obtained by analyzing public data from the Centers for Disease Control and Prevention (CDC) and Climate Prediction Center websites. It is evident that exposure of the eyes to UVB radiation upregulated *Pomc* expression in the pituitary gland. Then, an increased α-MSH level in the serum was detected and found to protect mice from allergic airway inflammation by reducing the function of ILC2s via the α-MSH receptor subtype *Mc5r*. A mechanistic study reveals that the α-MSH–MC5R signaling inhibited IL-7R-STAT and NF-κB pathways in activated pulmonary ILC2s, as evidenced by the downregulation of STAT3/5 and p65 phosphorylation. The therapeutic effects of α-MSH are validated with clinical data and mouse models of house dust mite (HDM)-induced and ovalbumin (OVA)-alum-induced chronic inflammation. We also note that plasma α-MSH concentrations were negatively correlated with the number and function of ILC2s among peripheral blood mononuclear cells (PBMCs) in patients with asthma. These findings provide insights into the role of neuroendocrine α-MSH in targeting ILC2s to limit lung inflammation and provide an option for controlling allergic lung inflammation. Our study also provides some scientific evidence to support century-old anecdotal reports that beach and mountain resort holidays associated with increased UV exposure through sunlight are beneficial in lung inflammation treatment.

## Results

### Solar UVB exposure attenuates asthma attacks in human patients and type 2 allergic mouse models

To investigate the relationship between UVB and asthma attacks, we analyzed data from a 5-year (2015–2019) asthma visit (unscheduled) survey of 18,8431 people in Shanghai. Minor daily asthma emergency department (ED) visits over 5 years were mainly concentrated in the summer, which had the highest monthly direct solar UVB radiation data (W/m²) (Fig. 1a, Supplementary Fig. 1a). The daily asthma ED visits showed a negative correlation with the daily direct solar UVB radiation

data (W/m²) ($R^2 = 0.1936$, $p$-value < 0.0001) (Fig. 1b), and both female and male asthma ED visits are affected by UVB (female: $R^2 = 0.2042$, $p$-value < 0.0001; male: $R^2 = 0.1324$, $p$-value < 0.0001) (Supplementary Fig. 1b). Additionally, we analyzed the relationship between asthma prevalence (data from the Centers for Disease Control and Prevention) and the corresponding UV index (data from the Climate Prediction Center) in all U.S. states between 2014 and 2020. The prevalence of asthma in states or regions with a higher cloudy UV index was lower than that in those with a lower cloudy UV index (Supplementary Fig. 1c). The average annual asthma prevalence was also negatively correlated with the average cloudy UV index ($R^2 = 0.1009$, $p$-value = 0.0246) (Supplementary Fig. 1d). Taken together, our data and the public data both indicate that asthma attacks may be negatively related to solar UV radiation.

Environmental factors, such as cold temperatures[34], air ozone[35], and air quality (PM2.5, PM10)[36,37], are linked with increased asthma attacks. The relationship between UV radiation and asthma prevalence in patients has been mentioned[38], but direct evidence is lacking, and the mechanism remains unclear. Acute lung allergic mouse models driven by ILC2s were used to confirm the direct cause-and-effect relationship between sunlight exposure and asthma attacks. Each mouse was treated with UVB exposure (2.5 kJ/m², equivalent to one minimal edema dose) before sensitization by papain administration for 5 days (Fig. 1c). UVB-irradiated mice showed reduced infiltration of eosinophils (Fig. 1d, e). Lung inflammation remission was detected by hematoxylin–eosin (H&E) and periodic acid–Schiff (PAS) staining of lung tissues (Fig. 1f). The levels of the type 2 cytokines IL-5 and IL-13 in the bronchoalveolar lavage fluid (BALF) (Fig. 1g) and the numbers of ILC2s in the lungs (Fig. 1h), as well as their capacity to produce the effector cytokines including IL-5 and IL-13(Fig. 1i, j), were consistently lower in UVB-irradiated mice than in nonirradiated controls.

Next, the adoptive transfer of ILC2s into Rag2⁻/⁻Il2rg⁻/⁻ mice, which lack T cells, B cells, NK cells, and ILCs, was performed. ILC2s from the lung of IL-33-treated mice were adoptively transferred into Rag2⁻/⁻Il2rg⁻/⁻ mice, followed by UVB radiation (Fig. 2a). UVB radiation reduced the infiltration of eosinophils (Fig. 2b), IL-5 and IL-13 production (Fig. 2c) and the numbers of eosinophils and ILC2s in the lung (Fig. 2d) after ILC2s transfer. Thus, these results indicate that UVB exposure may be beneficial in alleviating ILC2-mediated allergic lung inflammation.

To explore whether other cells are affected by UVB which in turn inhibits ILC2s (Supplementary Fig. 2a), we detected the differences among other cell types in the lung, such as neutrophils, macrophages (Supplementary Fig. 2b), T cells, B cells, natural killer cells (Supplementary Fig. 2c) and dendritic cells (Supplementary Fig. 2d) between nonirradiated and UVB-irradiated mice with papain stimulation. These results showed that eosinophils and ILC2s were reduced after UVB radiation (Supplementary Fig. 2e), and the other cell types remained unaffected (Supplementary Fig. 2f–h). These observations collectively suggested that UVB radiation may mainly suppress the ILC2 function.

Because ILC2 could be inhibited by sex steroid hormones such as androgens[28], both female mice and male mice were treated with UVB exposure before sensitization by papain administration to explore the effect of sex steroid hormones (Supplementary Fig. 3a). Consistent with previously published reports, the severity of lung inflammation and ILC2s maintenance in male mice were overall lower than in female mice. However, UVB could inhibit lung inflammation in both female and male mice, including eosinophils infiltration (Supplementary Fig. 3b, c), the levels of the type 2 cytokines IL-5 and IL-13 in the BALF (Supplementary Fig. 3d), and the numbers of ILC2s, IL5⁺ILC2s and IL-13⁺ILC2s in the lungs (Supplementary Fig. 3e, f).

The UVB response system involves corticotropin-releasing hormone (CRH) and precursor hormone proopiomelanocortin (POMC)-derived peptides, such as melanocyte-stimulating hormone (MSH) and

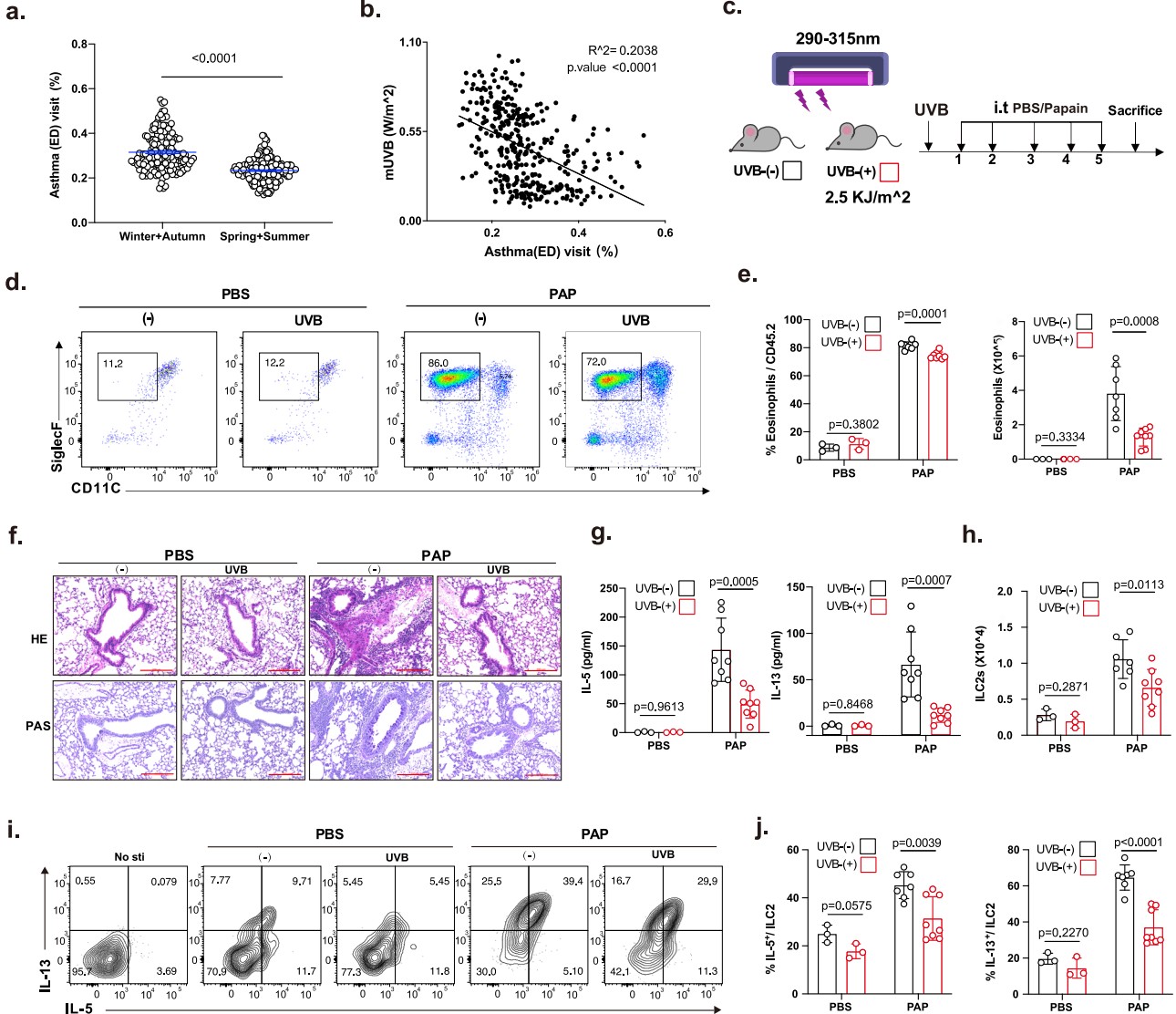

**Fig. 1 | Solar exposure decreased the asthma visit rate and suppressed ILC2-related lung inflammation. a** Daily average asthma (ED) visit distribution in winter + autumn (October–February) and spring + summer (March–September). **b** Correlation analysis between the average asthma visit rate (daily average) and corresponding UVB radiation from 2015 to 2019 in Shanghai. **c** WT female mice (6–8 weeks) were intratracheally challenged with papain (5 μg per mouse) or PBS for 5 consecutive days, and UVB radiation (2.5 kJ/m², equivalent to one minimal oedemal dose) exposure occurred 1 day before sensitization. The mice were sacrificed on day 6. **d** Flow cytometry plots for eosinophils among live CD45+ cells in the BALF of WT mice treated with or without UVB irradiation. **e** Percentage and number of eosinophils in the BALF (Eosinophils, Eos; FVD⁻CD45⁺CD11c⁻/ˡᵒSiglecF⁺) (n = 3, 3, 7, 8 from left to right). **f** Representative H&E and PAS staining of lung

sections (bars, 200 μm) (n = 3 in PBS group; n = 7 in papain group). **g** The levels of IL-5 and IL-13 in the BALF were measured by ELISA (n = 3,3,8,8 from left to right). **h** Statistical analysis of the numbers of ILC2s in the lungs (ILC2s, FVD⁻CD45.2⁺LIN⁻CD90.2⁺CD127⁺ST2⁺) (n = 3, 3, 7, 8 from left to right). **i** Flow cytometry plots for IL-5 and IL-13 in lung ILC2s stimulated with PMA plus ionomycin and BFA for 4 h. **j** Flow cytometric analysis of IL-5 and IL-13 in lung ILC2s stimulated with PMA plus ionomycin and BFA for 4 h (n = 3, 3, 7, 8 left to right). Each symbol represents an individual mouse (**e**, **g**, **h**, **j**). The bars and error bars show the means ± SDs. Data are representative of two or more independent experiments. For statistical analysis, the following tests were used. **b** Pearson's correlation coefficient analysis. **a**, **e**, **g**, **h**, **j** Two-tailed unpaired Student's t-test.

β-endorphin[39,40], which are released into the systemic circulation to stimulate receptor-expressing cells[41] (Fig. 2e). To explain the inhibitory function of UVB exposure in ILC2-mediated allergic inflammation, we speculated about the expression levels of the related receptors in ILC2s. RNA sequencing (RNA-seq) analysis revealed that murine ILC2s exhibited higher gene expression of *Mc5r* than the other α-MSH receptors (Fig. 2f), which was further confirmed by quantitative PCR in murine lung ILC2s (Fig. 2g). In addition, we demonstrated higher *Mc5r* expression in ILC2s than in other lung immune cell populations (Fig. 2h). The observation that the higher expression of MC5R in murine ILC2s indicated that this receptor may play a potential role in ILC2 response.

## UVB exposure in mice results in elevated α-MSH production through the hypothalamic–pituitary (HP) axis

MC5R is a member of five melanocortin receptors (termed MC1-5R), and its ligand is MSH. α-MSH is an endogenous tridecapeptide naturally generated from the precursor hormone proopiomelanocortin (POMC) that can be mainly detected in pituitary cells[42]. Two tissues are chronically exposed to UV radiation, including the skin and the eyes[43], and this exposure induces systemic production of α-MSH[41,44]. To compare the contributions of these two tissues, the shaved back skin of C57BL/6 mice (eyes covered) and the exposed eyes of C57BL/6 mice (back skin covered) were irradiated with 2.5 kJ/m² of UVB rays (Fig. 3a). The serum of mice with UVB radiation-exposed eyes showed a higher

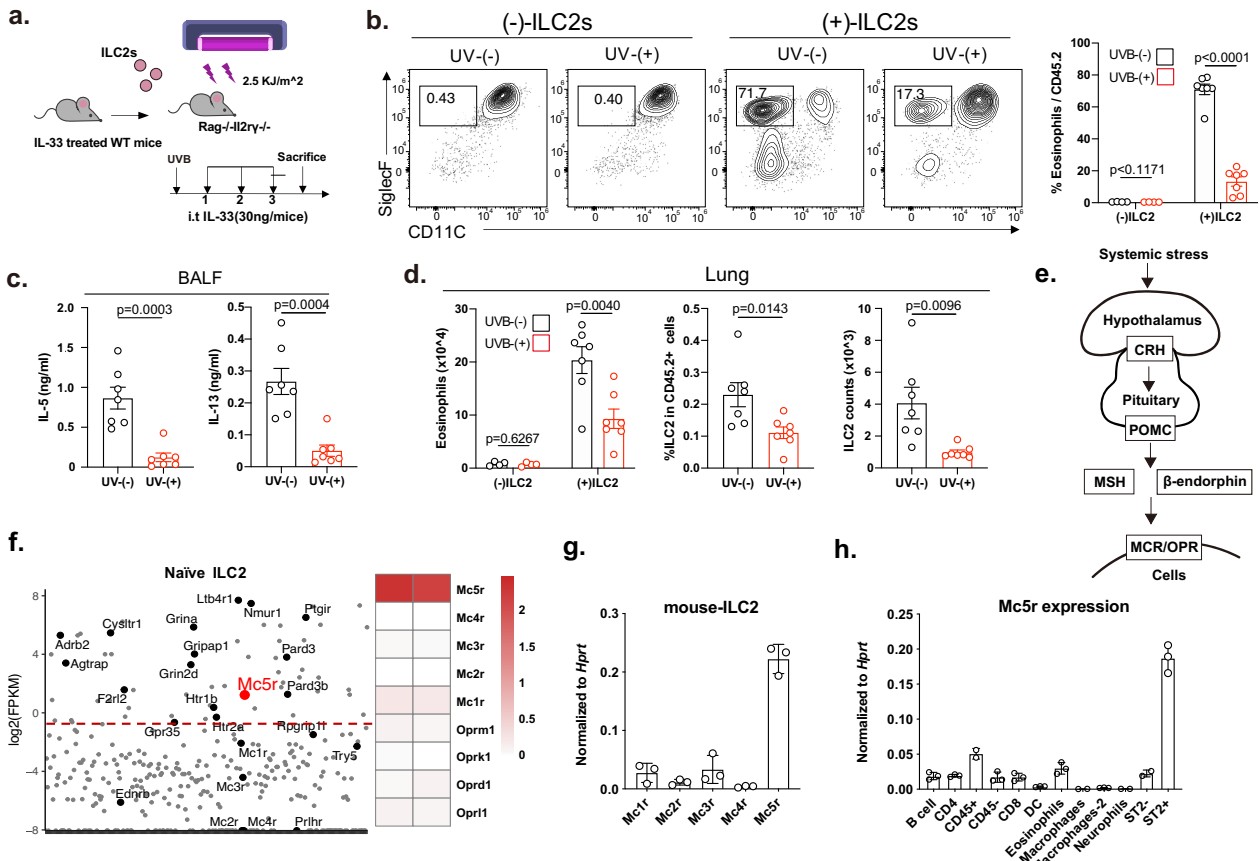

**Fig. 2 | *Mc5r* expression in mouse ILC2s. a–d** Adoptive transfer of ILC2s into Rag2$^{-/-}$Il2rg$^{-/-}$ female mice (6–8 weeks). Lung ILC2s ($4 \times 10^5$) from IL-33 treated mice were injected intravenously into Rag2$^{-/-}$Il2rg$^{-/-}$ recipients, followed by UVB radiation (2.5 kJ/m$^2$) before i.t. administrated with IL-33 (30 ng per mouse) for 3 days. Rag2$^{-/-}$Il2rg$^{-/-}$ mice without ILC2s transfer were used as controls. **a** The experimental strategy of ILC2 transfer. **b** Abundance of eosinophils in BALF of recipients ($n = 4, 4, 7, 7$ from left to right). **c** The levels of IL-5 and IL-13 in the BALF were measured by ELISA ($n = 7, 7$ from left to right). **d** Absolute counts of lung eosinophils ($n = 4, 4, 7, 7$ from left to right); Percentage and number of ILC2s from recipients upon ILC2 transfer and UVB radiation ($n = 7, 7$ from left to right). **e** Hypothalamic–pituitary axis. **f** Scatter plot showing the FPKM for nerve-related gene expression in lung ILC2s sorted from naïve mice. Mc-r and opioid receptor gene expression in lung ILC2s sorted from naïve mice and analyzed by RNA-seq are shown as a heatmap.

**g** Relative expression of *Mc-r* in sorted lung ILC2s ($n = 3$/group). **h** Relative expression of *Mc5r* in the indicated populations of sorted immune cells from the lungs (B cells: FVD$^-$CD3$^-$CD19$^+$; CD4$^+$ cells: FVD$^-$CD3$^+$CD4$^+$; CD45$^+$ cells: FVD$^-$CD45$^+$; CD8$^+$cells: FVD$^-$CD3$^+$CD8$^+$; DCs: FVD$^-$CD45$^+$CD103$^+$MHC-II$^+$CD11C$^+$; Eosinophils: FVD$^-$CD45$^+$CD11c$^{-/lo}$SiglecF$^+$; AMs alveolar macrophage, FVD$^-$CD45$^+$Ly6G$^-$CD11C$^+$SiglecF$^+$; interstitial macrophages IM: CD45$^+$Ly6G$^-$SiglecF$^-$CD11c$^+$CD11b$^+$F4/80$^+$; Neurophils: FVD$^-$CD45$^+$Ly6G$^+$CD11B$^+$; ILC2s: FVD$^-$CD45.2$^+$LIN$^-$CD90.2$^+$CD127$^+$ST2$^+$) ($n = 3$/group). Gene expression was analyzed by qRT-PCR and normalized with Hprt using the $2^{-\triangle\triangle Ct}$ method. Each symbol represents an individual mouse (**b–d, g, h**). The bars and error bars show the means ± SEMs. Data are representative of two or more independent experiments. For statistical analysis, the following tests were used. **b–d** Two-tailed unpaired Student's $t$-test.

level of α-MSH after irradiation (Fig. 3b). CRH expression in the hypothalamus increased at 12 h after UVB irradiation of the eyes at both the transcriptional (Fig. 3c) and protein levels (Fig. 3d, e). The expression of *Pomc*, which is highly expressed in the pituitary gland (Supplementary Fig. 4a, b), was increased in eyes-irradiated mice compared to sham-irradiated mice (Fig. 3f). Compared with the expression of other hormones (*Gh, Fshb, Prl,* and *Tshb*) in the pituitary gland and opioid precursors (*Pdyn, Penk,* and *Pnoc*), only the expression of *Pomc* was elevated in the eyes-irradiated mouse group (Supplementary Fig. 4c, d). PC1/3 (*Pcsk1*) cleaves POMC to produce β-lipotropin (β-LPH) and ACTH, whereas PC2 (*Pcsk2*) can further cleave ACTH and β-LPH to yield α-MSH and β-endorphin[39,40] (Fig. 3g). The expression of *Pcsk2*, but not that of *Pcsk1*, was elevated after UVB irradiation of the eyes (Fig. 3h). Consistent with this observation, UVB irradiation of the eyes increased the POMC protein levels in the pituitary gland (Fig. 3i, j). FACS (Supplementary Fig. 4e and Fig. 3k) and ELISA (Fig. 3l) analyses showed that α-MSH but not β-endorphin inhibited the functions of these ILC2s in vitro. The α-MSH release inhibitor was used in the mouse lung inflammation model

(Supplementary Fig. 5a), which reduced the circulating α-MSH levels in UVB-control mice (Supplementary Fig. 5b). Injection of the α-MSH release inhibitor could rescue the phenotype of lung inflammation, such as eosinophilic infiltration type 2 cytokine production, and lung damage shown by histological staining (Supplementary Fig. 5c–f). These data indicate that UVB radiation can activate the HP axis through a visual pathway, leading to increased α-MSH levels to restrain type 2 lung inflammation.

**UVB irradiation of the eyes protects against lung inflammation by reducing ILC2 function through the α-MSH−MC5R axis**

To investigate the contribution of MC5R in ILC2s in response to α-MSH treatment, we generated a genetically modified mouse strain that conditional deleted MC5R in ILC2s. Mice with Cre recombinase engineered under the control of the *Il-5* locus[8] were crossed with mice carrying *loxP* sites flanking the whole exon of the *MC5R* gene (Supplementary Fig. 6a–c). In the steady state, the percentage and number of ILC2s in various tissues were unperturbed in *Mc5r*$^{fl/fl}$R5$^{/+}$ mice (Supplementary Fig. 6d, e). Importantly, ILC2-dependent protection

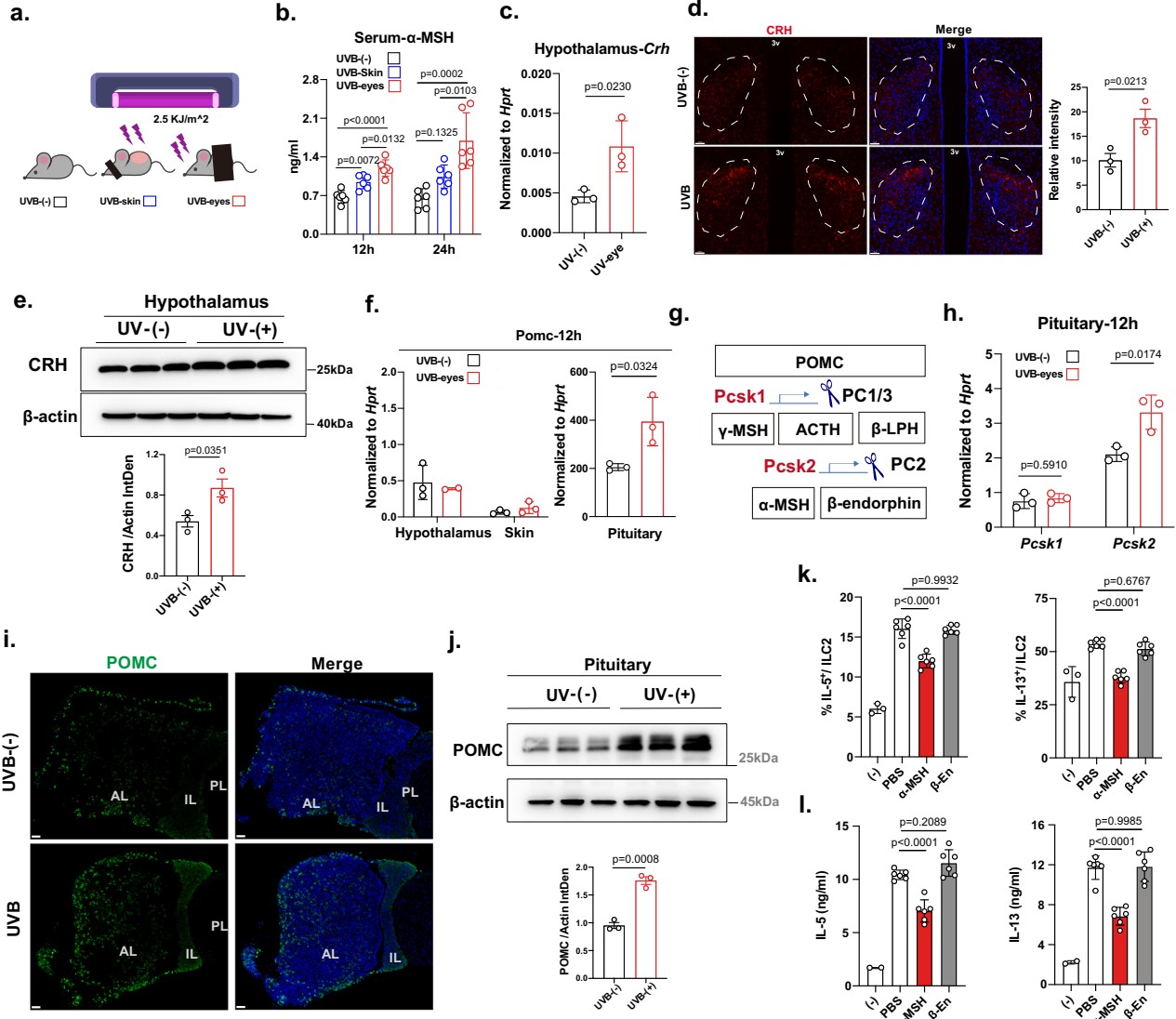

**Fig. 3 | UVB regulated the release of a-MSH by the neuroendocrine system to affect ILC2 function. a** Schematic diagram of the experimental design (6–8 weeks, female mice). **b** α-MSH in the serum were measured by ELISA (n = 6/group). **c** qRT–PCR analysis of *Crh* expression in the hypothalamus (n = 3/group). **d** Representative CRF staining in the hypothalamus (n = 3/group) (CRH: red; DAPI: blue; scale bar, 50 μm). Representative CRH quantification in the hypothalamus. **e** Proteins were comigrated by SDS–PAGE with homogenized hypothalamic tissue and visualized by Western blot analysis with anti-mouse CRH antibodies. Statistical analysis of CRH expression was performed using ImageJ (n = 3/group). **f** qRT–PCR analysis of *Pomc* expression in the hypothalamus, skin, and pituitary gland (n = 3, 2, 3, 3, 3 from left to right). **g** Melanocortin peptides (shaded boxes) derived from POMC. **h** qRT–PCR analysis of *Pcsk1* and *Pcsk2* expression in the pituitary gland (n = 3/group). **i** Representative POMC staining in the pituitary gland (POMC: green; DAPI: blue; scale bar, 50 μm). **j** Proteins were comigrated by SDS–PAGE with homogenized pituitary gland tissue and visualized by Western blot analysis with anti-mouse POMC antibodies. Statistical analysis of POMC expression was performed using ImageJ (n = 3/group). **k** Flow cytometric analysis of IL-5 and IL-13 in lung ILC2s 3 days after α-MSH/β-endorphin treatment stimulated with PMA plus ionomycin and BFA for 4 h (n = 3, 6, 6, 6 from left to right). **l** ELISA was performed to measure IL-5 and IL-13 in ILC2s culture medium after 3 days of treatment with α-MSH/β-endorphin was detected by ELISA (n = 2, 6, 6, 6 from left to right). Lung ILC2s sorted from IL-33-treated WT female mice were cultured in vitro with IL-2 (100 U/ml), mIL-7 (20 ng/ml), and mIL-33 (1 ng/ml) in the presence or absence of α-MSH/β-endorphin. Each symbol represents an individual mouse (**b**–**f**, **h**, **j**). Gene expression was analyzed by qRT–PCR and normalized with Hprt using the $2^{-\triangle\triangle Ct}$ method (**c**, **f**, **h**). The data are representative of at least three independent experiments. The bars and error bars show the means ± SDs. For statistical analysis, the following tests were used. **b**, **k**, **l** one-way ANOVA with Tukey's multiple comparisons test. **c**–**f**, **h**, **j** Two-tailed unpaired Student's t-test.

via UVB irradiation of the eyes in vivo was no longer observed in *Mc5r*$^{fl/fl}$R5$^{/+}$ mice (Fig. 4a) after we checked the effects on eosinophil accumulation (Fig. 4b), the degree of pulmonary inflammation (Fig. 4c, and Supplementary Fig. 7a), type 2 cytokine secretion (Fig. 4d), and the number and function of ILC2s (Fig. 4e, f). Consistent with previous results, α-MSH suppressed lung inflammation in R5$^{/+}$ mice but not in *Mc5r*$^{fl/fl}$R5$^{/+}$ mice (Fig. 4g–l, and Supplementary Fig. 7b). In addition to analyzing the differences between R5$^{/+}$ and *Mc5r*$^{fl/fl}$R5$^{/+}$ mice, we also re-performed the experiments to analyze the difference between the WT and KO groups under inflammatory conditions by using unpaired

Student's t-test. We found that conditional knockout of MC5R on ILC2 resulted in exacerbated lung inflammation and increased ILC2s response (Supplementary Fig. 8a–h).

The HPA axis also regulates the response to stress and various biological processes through the production of glucocorticoids (GCs) by the adrenal glands. GCs (cortisol in humans, corticosterone in rodents) regulate metabolism and immune response. To exclude the role of UVB-CGs on ILC2s, we measured the serum corticosterone in the UVB-skin mice (eyes covered) and UVB-eye mice (back skin covered). Consistent with the previous report[45], the exposure of the skin

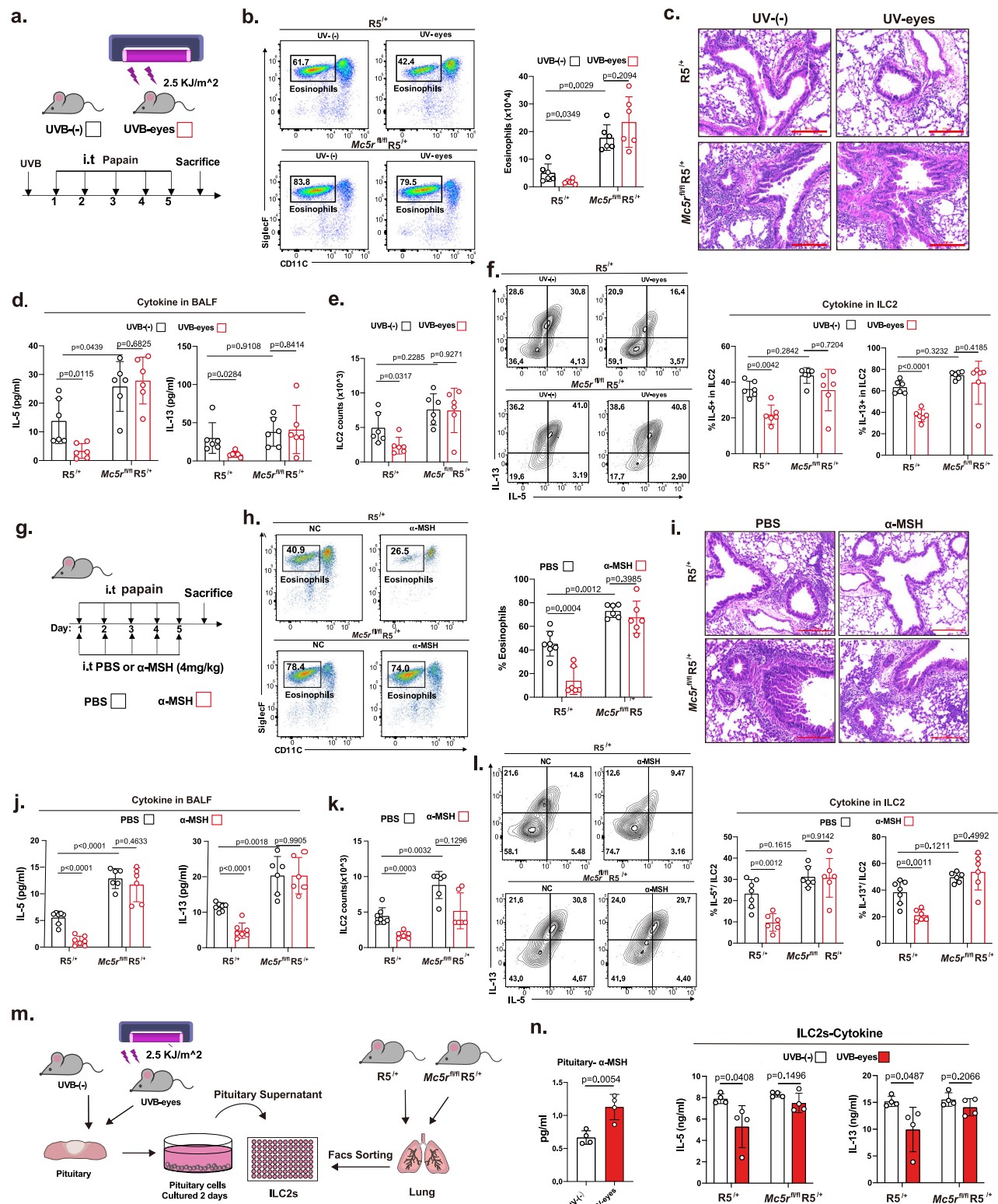

to UVB increased the levels of corticosterone. However, UVB exposure to the eye did not enhance the secretion of corticosterone in mice serum (Supplementary Fig. 9a). To compare the biological activity of endogenous α-MSH from the pituitary gland and corticosterone from the adrenal gland under the UVB-eyes irradiation condition, we cultured Mc5r-sufficient and Mc5r-deficient ILC2s in the supernatant of pituitary cells and adrenal gland cells cultures from wild-type mice with or without UVB irradiation to eyes (Fig. 4m, Supplementary Fig. 9b). The levels of α-MSH were higher in the culture supernatants of

mouse pituitary cells from mice with UVB-irradiated eyes (Fig. 4n, left). The levels of corticosterone did not increase significantly in the culture supernatants of the adrenal gland cells from the UVB-irradiated eyes group (Supplementary Fig. 9c). The UVB-mediated reduction of type 2 cytokines in ILC2s, which could be abolished after MC5R deletion, was only found in the pituitary cells group(Fig. 4n, right), but not in the adrenal gland cells group. (Supplementary Fig. 9d). These experiments indicate the UVB-eyes axis is more likely to play an inhibitory function in ILC2s through α-MSH/MC5R.

**Fig. 4 | UVB radiation inhibits ILC2-mediated lung inflammation via MC5R. a** R5[/]⁺ and *Mc5r*[fl/fl]R5[/]⁺ female mice (6–8 weeks) were intratracheally challenged with papain (4 μg/mouse) or PBS for 5 consecutive days with or without UVB irradiation of the eyes 24 h before day 1 and then sacrificed on day 6. **b** Percentage of eosinophils in the BALF (*n* = 6/group). **c** Representative H&E staining of lung sections (bars, 200 μm) (*n* = 6/group). **d** IL-5 and IL-13 in the BALF were detected by ELISA (*n* = 6/group). **e** Statistical analysis of the numbers of ILC2s in the lungs (*n* = 6/group). **f** Flow cytometric analysis of the percentages of IL-5⁺ and IL-13⁺ cells among lung ILC2s stimulated with PMA plus ionomycin and BFA for 4 h (*n* = 6/group). **g** WT female mice (6–8 weeks) were intratracheally challenged with papain (4 μg/mouse) or PBS for 5 consecutive days, treated with PBS or α-MSH (5 mg/kg) daily, and sacrificed on day 6. **h** Quantification of the eosinophil percentage in the BALF (*n* = 7, 6, 6, 6 from left to right). **i** Representative H&E staining of lung sections (bars, 200 μm) (*n* = 6/group). **j** ELISA was performed to measure the levels of IL-5 and IL-13 in the BALF (*n* = 7, 7, 6, 6 from left to right). **k** Statistical analysis of the numbers of ILC2s in the lungs (*n* = 7, 6, 6, 6 from left to right). **l** Flow cytometric analysis of IL-5 and IL-13 in lung ILC2s stimulated with PMA plus ionomycin and BFA for 4 h (*n* = 7, 6, 6, 6 from left to right). **m** ILC2s from R5[/]⁺ and *Mc5r*[fl/fl]R5[/]⁺ female mice cultured with the culture supernatant of pituitary cells from WT female mice(6-8w) treated with or without UVB irradiation to the eyes. **n** α-MSH in the pituitary cell culture supernatant was measured by ELISA (*n* = 4/group); IL-5 and IL-13 in the supernatants were detected by ELISA (*n* = 4/sample). Each symbol represents an individual mouse (**b**–**h**, **j**–**m**). The data are representative of at least three independent experiments. The bars and error bars show the means ± SDs. For statistical analysis, the following tests were used. **b**–**h**, **j**–**m** Two-tailed unpaired Student's *t*-test.

## The α-MSH−MC5R axis restrains ILC2 function via JAK/STAT and NF-κB signaling downregulation

To gain insights into the molecular mechanisms contributing to Mc5r-mediated lung ILC2 function, we performed RNA-seq analysis to compare the global transcriptome between Mc5r-deficient and Mc5r-sufficient lung ILC2s isolated from papain-treated mice. Principal component analysis (PCA) showed that the transcripts of Mc5r-deficient and Mc5r-sufficient lung ILC2s clustered separately (Supplementary Fig. 10a). Gene expression analysis identified 1466 significantly changed genes (fold change > 2, *p*-value < 0.05) in Mc5r-deficient ILC2s compared to Mc5r-sufficient ILC2s. Among these genes, 857 were upregulated, and 609 were downregulated (Supplementary Fig. 10b). Gene set enrichment analysis (GSEA) showed enrichment in asthma pathways (Fig. 5a), including increased IL-13, IL-4, and IL-5 expression (Fig. 5b). Notably, some known ILC2 regulators were altered in *Mc5r*-deficient lung ILC2s (Fig. 5c), especially *Il7r* (Fig. 5d). In addition, Kyoto Encyclopedia of Genes and Genomes (KEGG) enrichment analysis indicated that *Mc5r*-deficient ILC2s significantly upregulated cytokine-to-cytokine receptor interactions, especially the IL-7R signaling pathway (Supplementary Fig. 10c, d). Pathways activated downstream of IL-7R include the JAK/STAT, PI3K, and MAPK signaling pathways[46]. GSEA showed more enrichment in JAK/STAT signaling pathways than in other signaling pathways (Fig. 5e; Supplementary Fig. 9e). qRT–PCR confirmed the increased *Il7r* expression in *Mc5r*-deficient ILC2s (Supplementary Fig. 9f). Flow cytometric analysis showed that the *Il7r* (CD127) expression in ILC2s was reduced by α-MSH in R5[/]⁺ mice, which was abrogated in *Mc5r*[fl/fl]R5[/]⁺ mice (Fig. 5f). Furthermore, we found that the inhibitory effect of α-MSH on ILC2s was dependent on IL-7. When more IL-7 was added, a more inhibitory effect of α-MSH on ILC2s was observed (Fig. 5g). Lung ILC2s were sorted, cultured in a medium without any cytokines for 24 h, and then stimulated with IL-7 for 0, 10, and 60 min. Compared with the R5[/]⁺ group, the α-MSH-treated group showed decreased phosphorylation of JAK1/3-STAT3/5 activated by IL-7, but this reduction was not observed in the *Mc5r*[fl/fl]R5[/]⁺ group (Fig. 5h). In vivo, the phosphorylation of STAT3 and STAT5 in ILC2s from R5[/]⁺ mice was also inhibited by α-MSH treatment, but this effect was abrogated in *Mc5r*[fl/fl]R5[/]⁺ mice (Supplementary Fig. 10g, h). Similar results were obtained in vitro experiments (Supplementary Fig. 10i). These observations collectively suggested that the α-MSH−MC5R axis affects the function of ILC2 by attenuating the activity of IL-7/IL-7R and the associated downstream STAT signaling pathway. In addition to the IL-7/IL-7R signaling pathway, IL-33 plays a very important role in ILC2s activation, it would be important to explore the role of α-MSH in the cytokine activation pathway of ILC2s. We cultured ILC2s under different cytokine conditions in the context of α-MSH treatment. In the culture condition of IL-2, IL-7 or IL-33, respectively, the treatment of a-MSH did not significantly inhibit the function of ILC2s to release cytokines (Fig. 5i). However, the inhibitory function of α-MSH was effective under the condition of IL-7 plus IL-33 but not IL-2 plus IL-33 (Fig. 5j). In the presence of all three cytokines, the inhibitory function of a-MSH on

ILC2 was amplified (Fig. 5j). Under the corresponding conditions, the apoptosis and proliferation of ILC2 were not significantly affected by α-MSH (Supplementary Fig. 10j). We suspected that IL-33 could upregulate the expression of CD127 (Supplementary Fig. 10k), enhance the signal of IL-7/IL-7R, and thus amplify the inhibitory function of α-MSH. Meanwhile, the phosphorylated p65 (active forms) in NF-κB pathways were downregulated by α-MSH treatment in IL-7 plus IL-33 condition (Fig. 5k). These findings indicated that α-MSH serves as a potent inhibitory signal for JAK/STAT and NF-κB pathways to regulate the production of type 2 cytokines in lung ILC2s.

## Effects of a-MSH in acute and chronic pulmonary allergic mice models

We next sought to investigate whether α-MSH treatment affects the characteristics of other allergic mice models. IL-33 is one of the key activators of ILC2s[47–49]. Among mice that received rmIL-33 intratracheally (Supplementary Fig. 11a), α-MSH-treated mice had milder inflammation (Supplementary Fig. 11b–d), fewer ILC2s (Supplementary Fig. 11e) and lower ILC2 cytokine production (Supplementary Fig. 11f) in the lungs than control mice. A similar observation was also obtained from the fungal allergen *Alternaria alternata-induced* mouse model (Supplementary Fig. 11g–l). To better understand the possible clinical significance of α-MSH for allergy treatment, HDM-induced and OVA-alum-induced mouse models were used (Fig. 6a and Supplementary Fig. 12a). Administration of antigens significantly increased markers of lung inflammation in the isotype-treated group but not in the α-MSH-treated group; these markers included total IgE, HDM -IgE, and HDM-IgG1 in the plasma (Fig. 6b); lung resistance (Fig. 6c); eosinophil accumulation (Fig. 6d); type 2 cytokine production in the BALF (Fig. 6e); histological staining results (Fig. 6f); and the number and function of ILC2s (Fig. 6g, h). The numbers and activation of ST2⁺CD4⁺T cells remained unaffected by α-MSH treatment in the HDM model (Fig. 6i, j). Similar phenotypes were also observed in the OVA-alum-induced mouse models model (Supplementary Fig. 12b–f). These data suggested that α-MSH engagement should have an inhibitory function both in acute and chronic lung inflammation.

## α-MSH in asthmatic patients is negatively correlated with the activation of ILC2s in PBMCs

We next explored the relevance in human asthmatic patients about the effects of the α-MSH−MC5R axis on ILC2 responses and type 2 inflammation. *MC5R* was found to be highly expressed in human CRTH2⁺ ILC2s sorted from human PBMCs but not in other immune cells (Supplementary Fig. 13a and Fig. 7a). α-MSH also inhibited type 2 cytokine secretion in IL-33-activated human ILC2s sorted from PBMCs in vitro (Fig. 7b). In asthmatic patients, the α-MSH concentration in the plasma was negatively associated with the frequency and amount of human ILC2s in PBMCs (Fig. 7c). Our previous study found that ILC2s were the primary source of type 2 cytokines among PBMCs activated with IL-33[50]. We cultured PBMCs in vitro with IL-2, IL-7, and IL-33 for 5 days to assess IL-33-mediated type 2 cytokines in allergic patients.

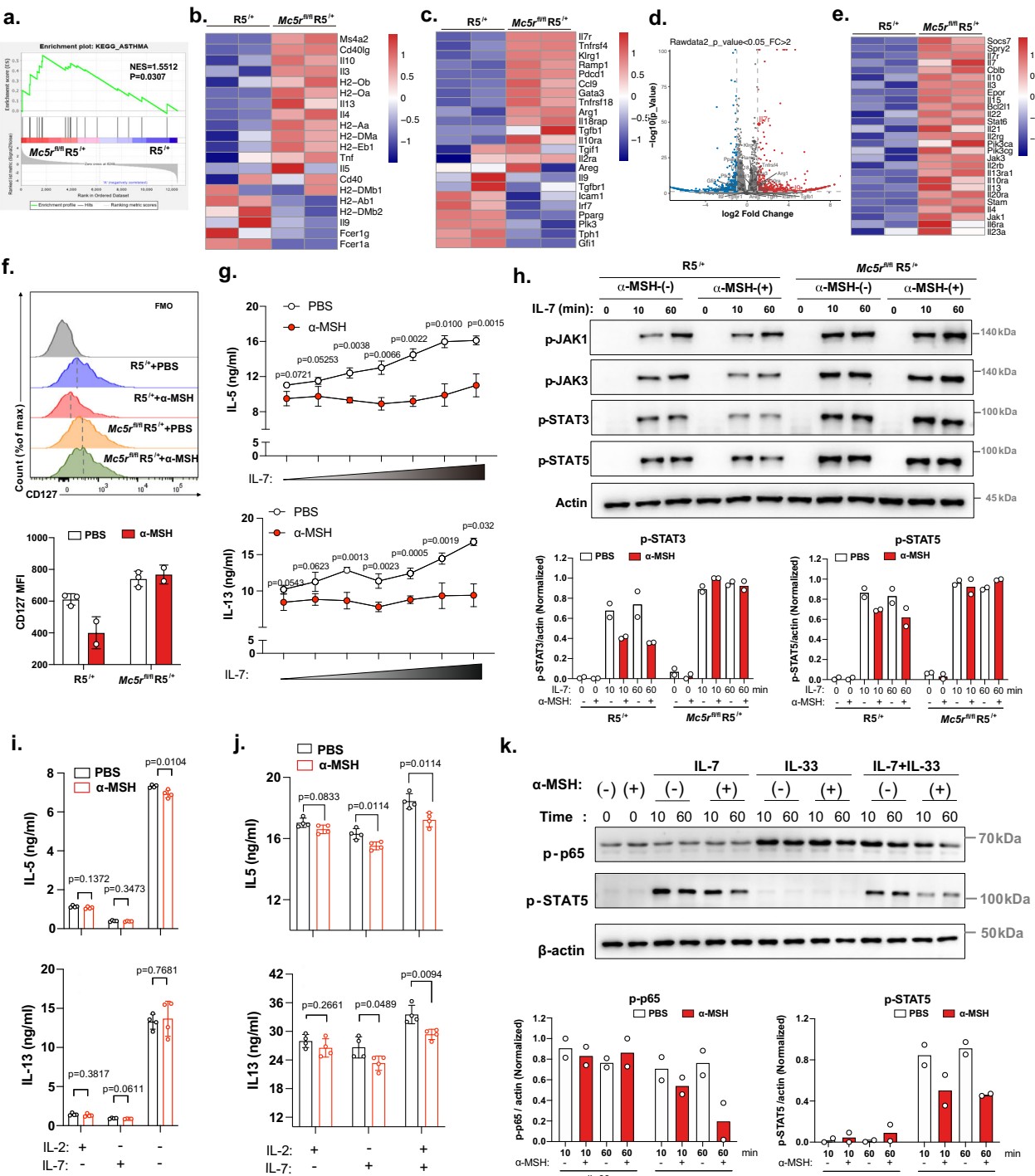

**Fig. 5 | JAK/STAT and NF-κB signaling are associated with the α-MSH−MC5R axis in ILC2s. a−e** Lung ILC2s were sorted from R5[/+] and *Mc5r[fl/fl]*R5[/+] female mice (6−8 weeks), and transcriptional profiling was performed. **a** GSEA of the asthma pathway. NES, normalized enrichment score. **b** Heatmap of asthma-characteristic genes in ILC2s. **c** Heatmap of known ILC2 regulators. **d** Volcano plot of differentially expressed genes (log2(fold change) > 2; *P* < 0.05) in ILC2s compared between R5[/+] and *Mc5r[fl/fl]*R5[/+] mice. The ILC2 regulators mentioned in D are highlighted. **e** Heatmap of selected genes in JAK/STAT signaling pathways. **f** MFIs of CD127 in ILC2s (*n* = 3, 2, 3, 2 from left to right). **g** ELISA was performed to measure IL-5 and IL-13 secretion into the mouse ILC2 culture medium after 3 days (*n* = 3/group). Lung ILC2s sorted from IL-33-treated WT female mice were cultured with IL-2 (100 U/ml), mIL-33 (1 ng/ml), and IL-7 (0, 2, 20, 200 pg/ml, 2, 20, and 200 ng/ml) ± α-MSH. **h** Sorted lLC2s were cultured ± α-MSH for 3 days, then cultured in the absence of

cytokines for 24 h and re-stimulated with IL-7 for 0, 10, and 60 min for Western blot analysis (up); the levels p-STAT3 and p-STAT5 were performed using ImageJ (down) (*n* = 2/group). **i** IL-5 and IL-13 in ILC2 culture medium were detected by ELISA after 3 days (*n* = 4/group). ILC2s cultured with IL-2 (100 U/ml), IL-7(20 ng/ml), or IL-33(1 ng/ml), respectively, ± α-MSH. **j** IL-5 and IL-13 were detected by ELISA after 3 days. ILC2s cultured in combinations of IL-2 (100 U/ml) plus IL-33(1 ng/ml), IL-7(20 ng/ml) plus IL-33(1 ng/ml), or IL-2 (100 U/ml) plus IL-7(20 ng/ml) and IL-33 (1 ng/ml) ± α-MSH (*n* = 4/group). **k** Sorted ILC2s were cultured ± α-MSH for 3 days, then cultured in the absence of cytokines for 24 h and re-stimulated with IL-7, IL-33, and IL-7 plus IL-33 for 0, 10, and 60 min for Western blot analysis (*n* = 2/group). The data are representative of two or more independent experiments. The bars and error bars show the means ± SDs. **g**, **i**, **j** Two-tailed unpaired Student's *t*-test.

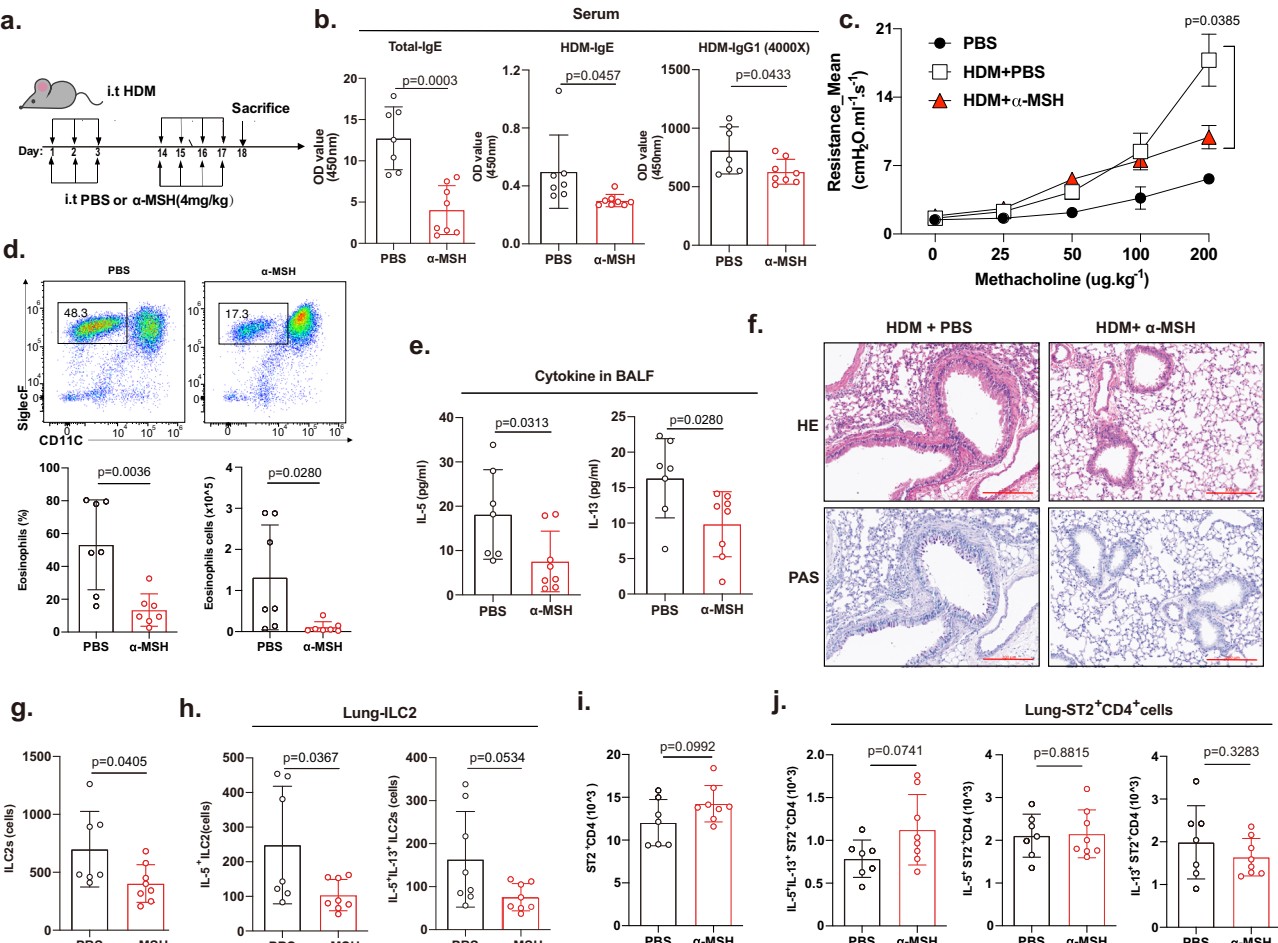

**Fig. 6 | The role of α-MSH in HDM-induced chronic lung inflammation. a** WT female mice (6–8 weeks) were challenged for 3 consecutive days with HDM (30 µg/mouse) and treated with PBS or α-MSH (4 mg/kg) daily. Then, the mice were allowed to rest for 10 days, rechallenged with HDM (6 µg/mouse) for 4 consecutive days, and sacrificed on day 18. **b** The total IgE levels, HDM-specific IgE, and HDM-specific IgG1 levels in the serum of PBS- and α-MSH-treated mice were measured by ELISA (n = 7, 8); the antibody titers were extrapolated from a standard curve (if available), or the results are presented as the absorbance values at 450 nm (OD450 values) for the serum dilution for which all samples were in the range of the assay. **c** Line graphs show lung resistance in response to increasing doses of methacholine (n = 2 mice for PBS, 4 mice for HDM, 4 mice for HDM+α-MSH). **d** Percentage and number of eosinophils in the BALF (n = 7/group). **e** The levels of IL-5 and IL-13 in the BALF were measured by ELISA (n = 7, 8). **f** Representative H&E and PAS staining of lung sections (bars, 200 µm) (n = 7, 8). **g** The number of ILC2s in the lungs (n = 7, 8). **h** The number of IL-5⁺IL-13⁺ cells among lung ILC2s after stimulation with PMA, ionomycin, and BFA for 4 h (n = 7, 8). **i** The number of ST2⁺CD4⁺T cells in the lungs (n = 7, 8). **j** The number of IL-5⁺IL-13⁺ cells among lung ST2⁺CD4⁺T cells after stimulation with PMA, ionomycin, and BFA for 4 h (n = 7, 8). Each symbol represents an individual mouse (**b**–**h**, **j**). The bars and error bars show the means ± SDs. **b**–**e**, **g**–**j** Two-tailed unpaired Student's *t*-test.

IL-13 and IL-5 secretion by PBMCs from patients with asthma were also negatively correlated with the plasma levels of α-MSH (Fig. 7d). In addition, type 2 cytokines were significantly reduced in an asthmatic PBMCs culture supernatant after in vitro treatment with a-MSH (Fig. 7e). In addition, there is no significant difference in the concentration of a-MSH between females and males (Supplementary Fig. 13b). All the patients' information is shown in Supplementary Fig. 13c. It would be valuable to explore whether α-MSH can intervene in ILC2-mediated type 2 immune response as a therapeutic approach. We evaluated the effect of α-MSH after the papain administration on the activation of ILC2s (Fig. 7f). Lung inflammation and ILC2 function were measured on day 10. Interestingly, the percentage and number of eosinophils were reduced in BALF from α-MSH-treated mice (Fig. 7g, h). α-MSH treatment also ameliorated inflammatory cell infiltration and mucus hypersecretion (Fig. 7i). Moreover, α-MSH-treated mice exhibited decreased ILC2 numbers in the lung (Fig. 7j). Intracellular staining of type 2 cytokines revealed a significant reduction in the percentage of IL-5- and IL-13-producing ILC2s (Fig. 7k, l). These observations collectively suggested that α-MSH has the potential in the treatment of ILC2-induced type 2 inflammatory diseases.

## Discussion

In this study, we uncovered a potential mechanism for pulmonary ILC2s in the context of allergic asthma, AHR, and lung inflammation and suggested the possibility of sunlight-assisted therapy. We found a negative correlation between solar UVB radiation and asthma attacks. We demonstrated that UVB radiation exposure of the eyes upregulated *Pomc* and *Pcsk2* expression in the pituitary gland in a mouse model and that these effects contributed to increased α-MSH production to lessen lung inflammation by restricting ILC2s via MC5R. The observed transcriptional profile and downregulation of IL-7R-JAK/STAT and p65 phosphorylation were consistent with these observations. The plasma α-MSH in asthmatic patients was negatively correlated with the response of ILC2s in PBMCs. These observations revealed that UVB radiation-driven neuroendocrine α-MSH is a regulator of lung ILC2s and raised a promising therapeutic avenue for lung inflammation treatment.

However, there are still several issues that remain to be discussed.
(1) Although we demonstrated that UVB irradiation of the eyes could affect the secretion of a-MSH from the pituitary gland, it is not clear which retinal signaling pathway is activated in the eyes by

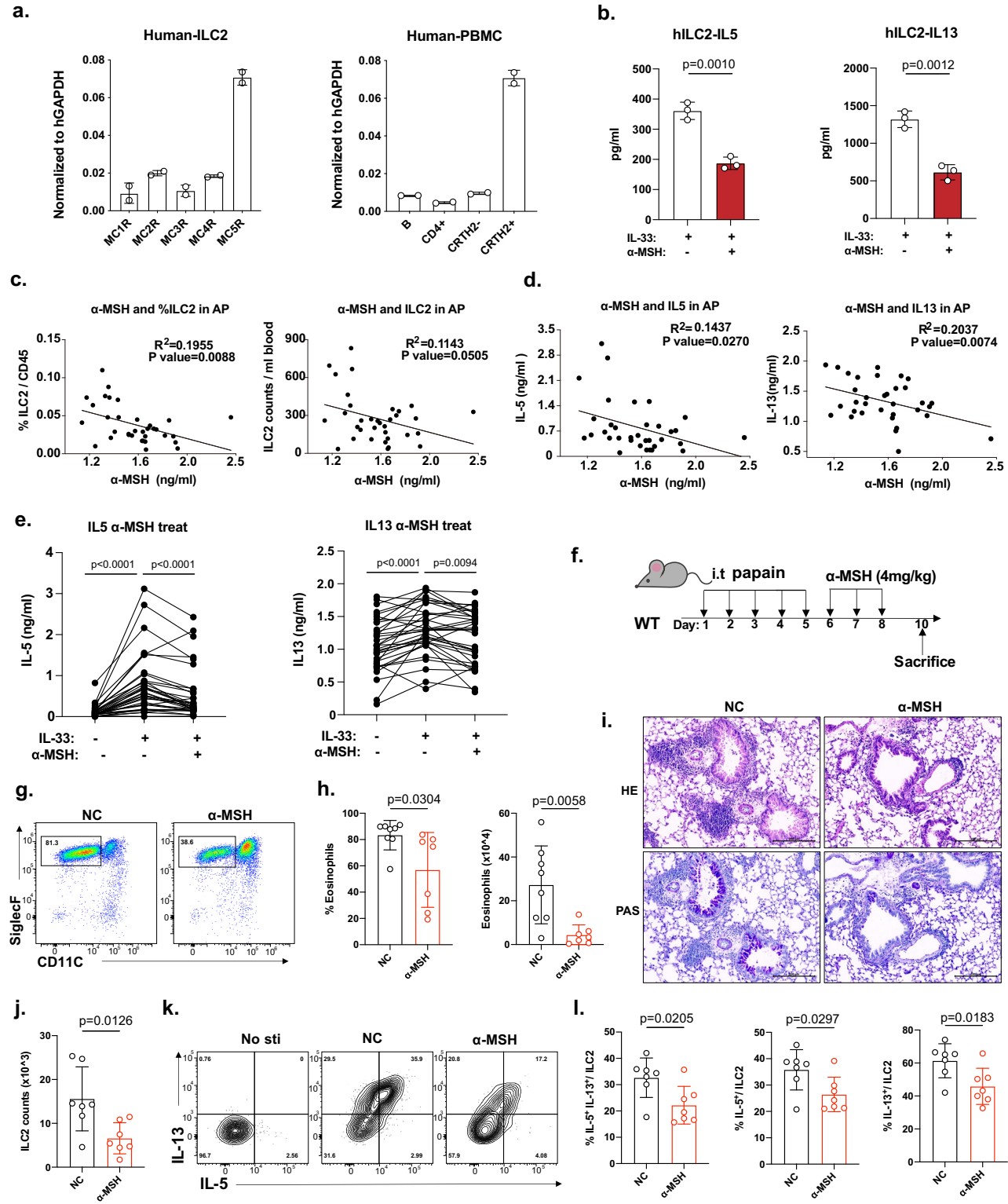

UVB radiation. A previous study showed that a signal induced by UVB irradiation of the eyes was transmitted through the ciliary ganglia involving the first branch of the trigeminal nerve to the hypothalamic-pituitary POMC system, resulting in the upregulation of α-MSH secretion[41]. Therefore, we hypothesized that the signal generated by UVB irradiation of the eyes is transmitted to the hypothalamic-pituitary system through a neuronal network that includes the ciliary ganglion to increase the systemic α-MSH concentration and affect ILC2 functions.

(2) The immunosuppressive effects of UVB radiation resulting from complex mechanisms, such as effects on cis-urocanic acid (cis-UCA)[51,52], nerve growth factor[53], glucocorticoids[45,54,55] and histamine[5]; reduced effector and memory T cell levels[56]; and effects on vitamin D3. Vitamin D production represents one of the most important beneficial effects of sunlight exposure[57]. But even so, some studies have suggested that vitamin D receptor deficiency fails to affect lung pathology[58]. Besides that, the transcription factor aryl hydrocarbon receptor (AhR) is a well-

**Fig. 7 | IL-33-mediated ILC2 activation in PBMCs from asthma patients (APs) is negatively associated with α-MSH levels. a** *MC5R* expression in hILC2s (FVD⁻CD45⁺LIN⁻CD161⁺CD127⁺CRTH2⁺) and CD4⁺ T cells and B cells sorted from human PBMCs (n = 2/group). *MC-R* expression in hILC2s sorted from PBMCs (n = 2/group). Gene expression was analyzed by qRT-PCR and normalized with hGAPDH using the $2^{-\triangle\triangle Ct}$ method. **b** hIL-5 and hIL-13 in hILC2s culture medium were detected by ELISA after 5 days of treatment with α-MSH (n = 3/group). hILC2s were cultured with hIL-2 (100 U/ml), hIL-7 (20 ng/ml), and hIL-33 (25 ng/ml) ±α-MSH. **c** Correlation analysis between the percentages and numbers of hILC2s among PBMCs and α-MSH concentration in the plasma of APs (n = 34). **d** Correlation analysis between IL-5/IL-13 and the α-MSH concentration in the plasma of APs (n = 34). **e** Secretion of IL-5 and IL-13 by PBMCs after IL-33 activation ± α-MSH (n = 34). PBMCs from APs cultured with hIL-2 (100 U/ml), hIL-7 (20 ng/ml), and hIL-33 (50 ng/ml). IL-5 and IL-13 in supernatants were measured by ELISA on day 5. PBMCs with one-fold higher IL-5

secretion after IL-2/IL-7/IL-33 treatment than after IL-2/IL-7 treatment were used for data analysis. **f** The WT female mice (6–8 weeks) were intratracheally challenged with papain for 5 consecutive days and intratracheally treated with PBS or α-MSH (4 mg/kg) for two consecutive days and sacrificed on day 10. **g** Flow diagram showing the percentage of eosinophils in the BALF. **h** Percentage and number of eosinophils in the BALF (n = 8, 7 from left to right). **i** HE and PAS staining of lung sections (bars, 200 μm) (n = 8, 7 from left to right). **j** Statistical analysis of the numbers of lung ILC2s (n = 7/group). **k** Flow cytometric analysis of IL-5 and IL-13 in lung ILC2s stimulated with PMA plus ionomycin and BFA for 4 h. **l** Statistical analysis of the percentages of IL-5⁺ and IL-13⁺ cells among lung ILC2s (n = 7/group). Each symbol represents an individual patient (**c**–**e**) or mouse (**h**, **j**–**l**). The bars and error bars show the means ± SDs. For statistical analysis, the following tests were used. **c**, **d** Pearson's correlation coefficient analysis. **b**, **h**, **j**, **l** Two-tailed unpaired Student's *t*-test. **e** one-way ANOVA with Tukey's multiple comparisons test.

known pleiotropic sensor of environmental factors, which can be activated by UVB radiation[59]. AhR has been demonstrated to inhibit ILC2 function[30], which raises another possibility that UVB may inhibit ILC2-mediated lung inflammation in a complex process orchestrating AhR activation and α-MSH/MC5R signaling. The UVB radiation–eyes–α-MSH–ILC2 axis we discovered might complement and improve the immunosuppressive mechanism of UVB radiation.

(3) It is widely acknowledged that ILC2s are an important target in several important diseases, such as parasitic intestinal infection, chronic obstructive pulmonary disorder (COPD), lung fibrosis, influenza infection, and atopic dermatitis[60]. In our study, we used only the most classic asthma models. Since UVB irradiation of the eyes causes a systemic increase in a-MSH levels, the role of the UVB radiation–eyes–α-MSH–ILC2 axis in other diseases needs to be further characterized.

Our studies described the inhibitory effects of UVB irradiation of the eyes on allergic disease in mice experiments. The amounts of UV exposure given were realistic and could be obtained with 10 min of summer sun exposure. With the mechanisms of UV irradiation better understood, we may be able to utilize the positive regulatory effects while avoiding the carcinogenic effects of UV exposure. Our studies in mice complement anecdotal reports that vacationing in the mountains (high altitude) or at the beach (torrid zone) with prolonged sun exposure is beneficial in reducing asthma symptoms. Sunlight exposure can also be considered a strategic health supplement, especially in "low doses", which are helpful beyond the role in vitamin D synthesis[61]. A more comprehensive analysis of human-environment interactions may lead to the development of therapeutics, which might benefit various pathologies and control many systemic immune diseases, not only allergic airway diseases.

Importantly, conditional deletion of the Mc5r gene led to higher ILC2 responses and severe lung inflammation in mice. These observations may be related to a-MSH autocrine under inflammatory conditions. Previous reports have demonstrated that asthmatic patients have elevated concentrations of α-MSH in BALF compared with healthy individuals[62]. Similarly, a similar phenomenon has been observed in experimental allergic mice[63]. Therefore, we hypothesized that during the inflammation of the lungs of the body, systemic production of α-MSH increased, and negatively regulates the aggravation of lung inflammation through MC5R on ILC2s, thereby maintaining the homeostasis of the airway inflammation. While studies on the developmental roles of IL-7/IL-7R signaling in ILC2 are substantial, research on its effector functions is relatively modest. IL-7 was recognized as an important factor in T cell development long before its role in effector function was examined. IL-7 is primarily produced in primary lymphoid tissues, such as bone marrow and thymic stromal cells, where immune cell development occurs[64,65]. However, ILC2s respond to multiple cytokines such as IL-33, IL-2, and IL-7, either individually or

synergistically to produce effector cytokines such as IL-5 and IL-13[9,47,48]. So multiple cytokines may have a synergistic role in promoting ILC2 activation.

Overall, our results suggest that the UVB radiation–α-MSH–MC5R axis provides an important inhibitory signal for the modulation of ILC2-induced lung inflammation and introduces α-MSH as a promising therapeutic target in allergic lung inflammation.

## Methods
### Ethics statement
The procedures in this study involving the collection of the average emergency asthma (ED) visit counts were approved by the Research Ethics Board of Ruijin Hospital, Shanghai Jiao Tong University School of Medicine. All animal experiments were approved by the Institutional Animal Care and Use Committee(IACUC) of the Shanghai Institute of Biochemistry and Cell Biology, Chinese Academy of Sciences. The collection and analysis of human PBMC in this study were approved by the Research Ethics Board of Ruijin Hospital, Shanghai Jiao Tong University School of Medicine. All manipulations were strictly conducted in compliance with animal ethics guidelines and approved protocols.

### Public data
The asthma prevalence data were obtained from the Centers for Disease Control and Prevention (https://www.cdc.gov/asthma/most_recent_data_states.htm) for 50 U.S. states from 2014 to 2020. The corresponding UV index (standardized by the UN's World Health Organization and World Meteorological Organization) data were obtained from the Climate Prediction Center (https://www.cpc.ncep.noaa.gov/products/stratosphere/uv_index/uv_annual.shtml).

### Mice
Wild-type mice, including those with C57BL/6 female and male mice, were purchased from the Shanghai Research Center for Model Organization (Shanghai, China). Rag1⁻/⁻ mice were maintained at the Chinese Academy of Sciences (Shanghai, China). *Mc5r*^fl/fl^ mice were purchased from GemPharmatech (Nanjing, China). Il5tm1.1 (icre)Lky/J(R5⁺/⁺) mice were ordered from Jackson laboratory. Mice were maintained at a macroenvironmental temperature of 21–22 °C, humidity (48–52%), in a conventional 12:12 light/dark cycle with lights on at 6:00 a.m. and off at 6:00 p.m. Mice were maintained under a specific pathogen-free conditions at the Animal Care Facility of the Chinese Academy of Sciences (Shanghai, China). We used 6-week-old to 8-week-old age-matched female and male mice in our studies. The animal care and use procedures complied with the guidelines of the Institute of Biochemistry and Cell Biology, Chinese Academy of Sciences. All animal experiments were approved by the Institutional Animal Care and Use Committee (IACUC) of the Shanghai Institute of Biochemistry and Cell Biology, Chinese Academy of Sciences (ethics approval no. SIBCB-S303-1610-030-c2). Mice were euthanized by $CO_2$ asphyxiation.

## Preparation of cell suspensions from lung tissue

Mice were euthanized, and BALF cells were collected from the lungs by gentle washing with 800 μl of ice-cold PBS twice with 18-G plastic cannulas and 1-ml syringes. For the isolation of lung cells, mice were perfused with 10 ml of cold DPBS via the right heart ventricle to remove blood from the lungs. Removed lungs were minced with scissors. Then, the minced lung tissue was placed in RPMI-1640 medium containing 10% fetal calf serum (FCS, Thermo Fisher) and 1% penicillin–streptomycin (Cytiva HyClone, SV30010) and digested with 0.5 mg/ml collagenase type I (Invitrogen, 17100017) for 30 min at 37 °C with continuous agitation in an incubator. The digested lung fragments were passed through a 70-μm filter to isolate the cell suspension and centrifuged at $350 \times g$ for 10 min. The red blood cells were lysed in BD Pharm lyse™ lysis buffer (BD Pharmingen, 555899). The supernatant was aspirated completely. The pellets were suspended in 40% Percoll (GE Healthcare, 17089109), which was layered over 80% Percoll; the gradient was centrifuged at $600 \times g$ for 20 min to obtain lung mononuclear cells, and the isolated cells were washed with RPMI-1640 medium before further analysis.

## Flow cytometric analysis and cell sorting

The following flow cytometry antibodies were used: antibodies specific for mouse CD45R-APC (RA3-6B2, 103212, 1/100), CD5-APC (53-7.3, 100626, 1/100), NK1.1-APC (PK136, 108710, 1/100), TCR γ/δ-APC (UC7-13D5, 118116, 1/100), Gr-1-APC (RB6-8C5, 108412, 1/100), the mouse erythroid cell marker TER119-APC (TER-119, 116212, 1/100), CD45.2-Pacific Blue (104, 109820, 1/150), Thy1.2- PerCP/Cyanine5.5 (53-2.1, 140322, 1/150), CD25-Brilliant Violet 421 (PC61, 102034, 1/100), IL-5-Brilliant Violet 421 (TRFK5, 504311, 1/100), and p-STAT3-Brilliant Violet 421 (13A3-1, 651010, 1/100) were purchased from BioLegend; antibodies specific to mouse CD3-APC (2C11, 17-0031-83, 1/100), CD11b-APC (M1/70, 17-0112-83, 1/200), CD11c-APC (N418, 17-0114-82, 1/400), FcεR1-APC (MAR-1, 17-5898-82, 1/200), IL-13-PE (eBio13A, 12-7133-82, 1/100), KLRG1- Pacific Blue (2F1, 48-5893-82, 1/100), SCA-1-PE-Cyanine7 (D7, 25-5981-81, 1/100), C-KIT-PerCP-eFluor™ 710 (2BB, 46-1171-82, 1/100), and CD16/CD32 (93, 14-0161-86, 1/100) were purchased from eBioscience; antibodies specific to CD127-PE-cy7 (A7R34, 560733, 1/100), Siglec-F-APC (E50-2440, 562680, 1/100), and p-p-STAT5-Alexa Fluor 647 (47/Stat5, 562076, 1/100) were purchased from BD Pharmingen; and antibodies specific to IL-33R-FITC (DJ8, 101001F, 1/100) were purchased from MD Bioproducts. Antibodies specific to human CD127-PE (A019D5, 351304, 1/100), CD161-APC/Cyanine7 (HP-3G10, 339928, 1/100), CRTH2-FITC (BM16, 350108, 1/100), and CD45-PerCP/Cyanine5.5 (HI30, 304028, 1/100) were purchased from BioLegend, and human Lineage Antibody Cocktail (8804-6836-74) was purchased from Thermo Fisher. A fixable viability stain (BD Pharmingen, 565388,1/1000) was used to discriminate viable cells from nonviable cells. For intranuclear staining of transcription factors, nuclear protein cytokines, or phosphorylated sites, cells were first stained with antibodies against surface antigens and then fixed and permeabilized according to the manufacturer's instructions (Foxp3/Transcription Factor Staining Buffer Set; eBioscience, 00-5223-56, 00-8333-56). For staining of phosphorylated proteins in sorted ILC2s, cells were fixed with Fixation Buffer (BioLegend, 420801) and permeabilized with True-Phos Perm Buffer (BioLegend, 425401) according to the manufacturer's instructions. For intracellular cytokine staining, cells were stimulated with 50 ng/ml phorbol 12-myristate 13-acetate (PMA; Merck Millipore, 524400), 1 μg/ml ionomycin (Merck Millipore, 407952), and 1 μg/ml brefeldin A (BFA; eBioscience, 00-4506-51) for 4 h. Flow cytometry data were collected on an LSR Fortessa (BD Pharmingen) or CytoFLEX3 flow cytometer (Beckman Coulter), and the data were analyzed with FlowJo V10 (FlowJo). For flow cytometric sorting, mouse lung ILC2s (CD45+Lin−CD127+CD90.2+ST2+) were sorted with an Aria Fusion or Aria III (BD Biosciences).

## Lung inflammation mouse models

To establish papain-induced acute type 2 airway inflammation, mice were anaesthetized and then intratracheally administered papain (4–5 μg per mouse; Sigma–Aldrich, P3125) for 5 consecutive days. Twenty-four hours after the final treatment, the mice were sacrificed, and samples were collected. For induction of HDM-induced chronic airway inflammation, mice were challenged with HDM (30 μg HDM per mouse; Greerlabs, XPB82D3A25) for 3 consecutive days. Then, we allowed the mice to rest for 10 days before rechallenging them with HDM (6 μg per mouse) for 4 consecutive days. The mice were sacrificed 24 h after the last challenge, and the BALF and lungs were collected for analysis. OVA (Sigma–Aldrich, A5503) was adsorbed onto Imject alum adjuvant (Thermo Fisher Scientific, 77161) by gentle shaking. Mice were primed by intraperitoneal injection of 20 μg of OVA-alum in a 200-μl volume. On day 14, the mice were 'boosted' with the same protocol. On days 21–25, the mice were challenged with aerosolized OVA (10 mg/ml) in PBS. On day 26, the mice were sacrificed for analysis. α-MSH (Abcam, ab120205, 4 mg/kg) or an α-MSH release inhibitor (Tocris, 1929, 1 mg/kg) was intratracheally (i.t.) or intraperitoneally (i.p.) administered according to the experimental design. The levels of α-MSH in the mice serum were measured by ELISA(Abmart, AB-J0275A).

## ILC2 culture

Fresh lung ILC2s (CD45+Lin−CD127+CD90.2+ST2+) sorted from IL33-treated mice were cultured in 96-well round-bottom plates at a density of $5 \times 10^3$ cells per well in RPMI-1640 medium containing 10% FCS, 100 U/ml penicillin and 100 mg/ml streptomycin in the presence of 10 ng/ml human IL-2 (PeproTech), 20 ng/ml mouse IL-7 (PeproTech, 217-17-100), 1 ng/ml mouse IL-33 (BioLegend, 580506), 20 ng/ml human IL-7 (R&D Systems, 207-IL-010) and 50 ng/ml human IL-33 (R&D Systems, 3625-IL-010) with or without α-MSH (10 ng/ml) or β-endorphin (MCE, HY-P1502, 10 ng/ml). Then, the levels of cytokines in the culture supernatants were measured by ELISA (mouse IL-5 ELISA kit, Invitrogen, 88-7054-76; mouse IL-13 ELISA kit, Invitrogen, 88-7137-76; human IL-5 ELISA kit, Invitrogen, 88-7056-88; and human IL-13 ELISA kit, Invitrogen, 88-7439-88). For intracellular cytokine staining, 50 ng/ml PMA, 1 μg/ml ionomycin, and 1 μg/ml BFA were added to the cultures 2–4 h before staining.

## ELISA

IL-5 and IL-13 were analyzed by ELISA as recommended by the manufacturer (Invitrogen). Total serum IgE was measured with the following reagents: anti-mouse IgE (BD pharmingen, capture antibody; 553413; 1/250), mouse IgE standard (SouthernBiotech; 0114-01), and Goat Anti-Mouse IgE-HRP (SouthernBiotech detection antibody; 1110-05; 1/16000). HDM and OVA-specific antibodies were analyzed by ELISA in plates that had been coated 3 h at 4 °C overnight with 100 μl HDM (50 μg/ml in PBS) or OVA (50 μg/ml in PBS), with nonspecific binding blocked by incubation for 1 h with 5% BSA. Wells were washed for three times and mouse serum samples diluted in assay diluent were added and incubated for 2 h. Wells were washed three times, then HRP Rat Anti-Mouse IgG1 (BD pharmingen; 559626; 1/250) or Goat Anti-Mouse IgG2c-HRP (SouthernBiotech; 1078-05; 1/4000) was added and incubated for 1 h. Wells were washed five times, and then were incubated with tetramethylbenzidine substrate until the reaction approached saturation. Linear regression analysis or measurement of absorbance was used to determine antibody titers or relative differences.

## Asthma ED visit surveys

The average emergency asthma (ED) visit counts for each day between 2015 and 2019 were obtained from Ruijin Hospital, Huashan Hospital, Dongfang Hospital, Shanghai Fifth People's Hospital, Shanghai Putuo District Central Hospital, Shanghai Putuo District People's Hospital, and Tongji Hospital. This study was approved by the Research Ethics Board of Ruijin Hospital, Shanghai Jiao Tong University School of

Medicine (ethics approval no. 2022-187). The corresponding UVB exposure data were obtained from the Shanghai Meteorological Service.

## Human samples

Asthmatic patients were recruited from Ruijin Hospital, Shanghai Jiaotong University School of Medicine. The diagnosis of asthma was established based on respiratory symptoms and evidence of variable airflow obstruction according to the Global Initiative for Asthma (GINA) guidelines. A lung function test was performed, and the predicted forced expiratory volume in 1 s (FEV1) percent (FEV1% pre), FEV1/forced vital capacity percentage (FEV1/FVC), disease duration and asthma control questionnaire (ACQ) scores were recorded. Patients with an acute attack, autoimmune diseases, or infectious diseases and those who had received systemic glucocorticoids and immunosuppressive agents within 1 month before the study were excluded. This study was approved by the Research Ethics Board of Ruijin Hospital, Shanghai Jiao Tong University School of Medicine (ethics approval no.2021-63). The levels of α-MSH in the patient's plasma were measured by ELISA(Abmart, AB-3769A). All patients who provided informed consent had samples collected; all study procedures were conducted in strict compliance with ethical and institutional regulations.

## Immunofluorescence (IF) staining

Formalin-fixed tissue sections were deparaffinized in xylene (10 min, twice) and then rehydrated in graded (100%, 95%, 85%, and 70%) alcohol solutions. The samples were processed for antigen retrieval in antigen retrieval buffer (EDTA, pH 9.0) at 100 °C for 15 min and then cooled slowly. After blocking with 10% BSA for 30 min, the sections were stained. The following antibodies were used: rabbit anti-CRH (Abcam, ab272391, 1/200) and rabbit anti-POMC (Abcam, ab210605, 1/500), followed by staining with goat anti-rabbit CY3 (Servicebio, GB21303, 1/300) and DAPI (Servicebio, G1012). Primary antibody staining was performed at 4 °C overnight. Secondary antibody staining was performed at room temperature for 50 min. Nuclei were stained with DAPI for 10 min. Images of representative tissue were acquired with a Pannoramic MIDI (3DHISTECH) and analyzed with CaseViewer 2.4.0.119028 software.

## Immunoblot analysis

Tissue was lysed in RIPA buffer (Beyotime) containing protein inhibitors (Roche). Protein concentration was determined by BCA protein assay (ShareBio). Proteins were mixed with SDS Page loading buffer (Beyotime) and incubated at 100 °C for 10 min. ILC2s ($1 \times 10^5$ cells) were lysed in 50 μl $1 \times$ SDS loading buffer containing phosphatase inhibitors (Roche). Then, protein lysate per lane was run through Gels and transferred to NC Membrane (Merck Millipore). The membrane was blocked for 1 h in 5% nonfat dried milk in Tris-buffered saline containing 0.1% Tween 20 (TBST) and incubated overnight with primary antibody at 4 °C. The membrane was then washed 3 times in TBST and incubated with an HRP-conjugated secondary antibody for 1 h at room temperature. Detection was performed with ECL western blotting detection reagents (ShareBio). The following antibodies were used: Rabbit anti-mouse CRH(Abcam,ab184238,1/5000); Rabbit anti-mouse POMC(Abcam, ab210605, 1/5000); Rabbit anti-mouse p-STAT3 (CST,9131s,1/1000); Rabbit anti-mouse p-STAT5 (CST,9359s,1/1000); Rabbit anti-mouse p-p65 (CST,3033s,1/2000); Beta Actin Monoclonal antibody (Proteintech, 66009-1-Ig,1/5000); Goat Anti-Rabbit IgG HRP Affinity Purified PAb (R&D Systems, HAF008,1/1000); Goat Anti-mouse IgG HRP Affinity Purified PAb (R&D Systems, HAF007,1/1000).

## Human PBMC collection and culture

We collected fresh blood in EDTA anticoagulant tubes at Shanghai Ruijin Hospital, Shanghai Jiaotong University School of Medicine. Human PBMCs were obtained by using Ficoll-Paque PLUS (GE Healthcare). The isolated PBMCs were washed and resuspended in PBS containing 2% fetal bovine serum (FBS). The cells were cultured in RPMI-1640 medium supplemented with hIL-2 (R&D Systems), hIL-7 (R&D Systems), and hIL-33 (R&D Systems). The IL-13 and IL-5 levels in the supernatant were then measured.

## Histological analysis

Mice were euthanized by $CO_2$ asphyxiation. The lungs were fixed for at least 24 h with 4% formalin (paraformaldehyde), embedded in paraffin, cut into 4-μm sections, and stained with H&E and Alcian blue (AB)/PAS according to standard protocols. The sections were scanned to evaluate inflammation using a light microscope. Images of H&E- and AB/PAS-stained tissue slides were acquired with a Zeiss Scan.Z1.2.3. Histologic images were analyzed with the ImageJ Analysis Application.

## Bulk RNA-seq analysis

In total, 5000-20000 ILC2s from each condition were lysed in TRIzolTM Reagent (Invitrogen). Libraries were constructed with SMART-Seq2 with two replicates for each condition. Total RNA was prepared, and RNA quality was assessed on an Agilent 2100 Bioanalyzer (Agilent Technologies). Total RNA was subjected to reverse transcription and amplification using the Single Cell Full Length mRNA-Amplification Kit (Vazyme, N712-02). A cDNA library was constructed with the TruePrep DNA Library Prep Kit V2 for Illumina (Vazyme, TD503-01) according to the manufacturer's protocol. Sequencing was performed on an Illumina NovaSeq 6000 platform. RNA-seq reads were mapped to the mouse genome GRCm38 with Hisat2 version 2.0.5 using default parameters. FeatureCounts v1.5.0-p3 was used to count the reads numbers. The differential expression was carried out with DESeq2 version 1.16.1 using default parameters.

## qPCR with reverse transcription

Total RNA was extracted from cells and tissues with TRIzol reagent (Takara) according to the manufacturer's instructions. RNA (1 μg) was reverse transcribed into cDNA with a synthesis kit (Vazyme, R323-01). Gene expression was analyzed by qRT–PCR with SYBR Green (Vazyme, Q712-03) on a Q6/384 system (Thermo Fisher Scientific). The sequences of the primers are listed. The relative expression of target genes was normalized to the expression of the gene encoding *mHprt* or *hGAPDH*. The relative mRNA expression of other genes was measured using the standard $2^{(-\triangle\triangle CT)}$ method. The sequences of primers were as follows: (5′ → 3′):

*mHprt*-forward: CTCATGGACTGATTATGGACAGGAC;
*mHprt*-reverse: GCAGGTCAGCAAAGAACTTATAGCC;
*mMc1r*-forward: AGAGCCTTGGTGCCTGTATG;
*mMc1r*-reverse: TGACACTTACCATCAGGTCAGAC;
*mMc2r*-forward: ACACCGCAAGAAATAACTCCG;
*mMc2r*-reverse: AGGAGGACAATCAAGTTCTCCA;
*mMc3r*-forward: TCCGATGCTGCCTAACCTCT;
*mMc3r*-reverse: GGATGTTTTCCATCAGACTGACG;
*mMc4r*-forward: CCCGGACGGAGGATGCTAT;
*mMc4r*-reverse: TCGCCACGATCACTAGAATGT;
*mMc5r*-forward: AGCCCGGTAAACAGAAGATTCA;
*mMc5r*-reverse: CTCTGAGGCGTTCAGGGTAAG;
*mCrh*-forward: CCTCAGCCGGTTCTGATCC;
*mCrh*-reverse: AGCAACACGCGGAAAAAGTTA;
*mPomc*-forward: TTCAGACCTCCATAGATGTG;
*mPomc*-reverse: GAAGTGACCCATGACGTA;
*mPenk*-forward: GAGAGCACCAACAATGACGAA;
*mPenk*-reverse: TCTTCTGGTAGTCCATCCACC;
*mPdyn*-forward: CTCCTCGTGATGCCCTCTAAT;
*mPdyn*-reverse: AGGGAGCAAATCAGGGGGT;
*mPnoc*-forward: TTCAACTTAAAGACGTGCATCCT;
*mPnoc*-reverse: GACACGCGGCATTCTCTTC;

*mGh*-forward: GCTACAGACTCTCGGACCTC;
*mGh*-reverse: CGGAGCACAGCATTAGAAAACAG;
*mTshb*-forward: GGGCAAGCAGCATCCTTTTG;
*mTshb*-reverse: GTGTCATACAATACCCAGCACAG;
*mFshb*-forward: GCCATAGCTGTGAATTGACCA;
*mFshb*-reverse: AGATCCCTAGTGTAGCAGTAGC;
*mPrl*-forward: CTGGCTACACCTGAAGACAAGG;
*mPrl*-reverse: TCACTCGAGGACTGCACCAAAC;
*mPcsk1*-forward: CTTTCGCCTTCTTTTGCGTTT;
*mPcsk1*-reverse: TCCGCCGCCCATTCATTAAC;
*mPcsk2*-forward: AGAGAGACCCCAGGATAAAGATG;
*mPcsk2*-reverse: CTTGCCCAGTGTTGAACAGGT;
*mIl7r*-forward: GCGGACGATCACTCCTTCTG;
*mIl7r*-reverse: AGCCCCACATATTTGAAATTCCA;
*hGAPDH*-forward: GTCAAGGCTGAGAACGGGAA;
*hGAPDH*-reverse: AAATGAGCCCCAGCCTTCTC;
*hMC1R*-forward: CCGCTACATCTCCATCTT;
*hMC1R*-reverse: GTCGTAGTAGGCGATGAA;
*hMC2R*-forward: GAAGCACATTATCAACTCGTA;
*hMC2R*-reverse: GCCTGGAGATTCTTATTCTTG;
*hMC3R*-forward: TTGTCTTTCCTGTGAGCA;
*hMC3R*-reverse: AGCACGAAGCATTCATTG;
*hMC4R*-forward: CCAAGGTGCCAATATGAAG;
*hMC4R*-reverse: TGAGGACAAGAGATGTAGAAT;
*hMC5R*-forward: TTGGATCTCAACCTGAATGCC;
*hMC5R*-reverse: GCCCCTATGACCAAGATGTTCTC.

### Induction and measurement of airway hyperreactivity
Mice were challenged with HDM (Greerlabs, XPB82D3A25) as shown in the experimental schemes. Lung function was measured by direct measurement of dynamic compliance in anaesthetized tracheostomized mice, which were mechanically ventilated with an AniRes2005 lung function system (Bestlab, version 3.5, China) and sequentially challenged with increasing aerosolized doses of methacholine (0, 25, 50, 100, and 200 µg/kg).

### Statistical analysis
Data are presented as the mean ± standard error of the mean (SEM) or standard deviation (SD). A two-tailed Student's *t*-test for unpaired data was applied for comparisons between the two groups. For multigroup comparisons, we used one-way ANOVA with the Tukey post hoc test. Data were analyzed with Prism software (GraphPad Prism 8). Error bars represent the SEM or SD.

### Reporting summary
Further information on research design is available in the Nature Portfolio Reporting Summary linked to this article.

## Data availability
Sequence data that support the findings of this study have been deposited in Genbank under the primary accession codes GSE237815 and GSE237962. All other data are available in the article and supplementary files or from the corresponding authors upon request. Source data are provided with this paper.

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

## Acknowledgements

We thank Dr. J. Qiu, Dr. J. Zhou, Dr. H. Wang, and Dr. J. Zhu for providing valuable insights into writing the manuscript. We are grateful to G. Lin for the animal breeding and management. We also acknowledge the individuals who provided technical support at the Core Facility for Cell Biology and the Animal Core Facility. This work was supported by the National Natural Science Foundation of China (32000667), Shanghai Science and Technology Innovation Action (21JC1405800, 20S11901800 and 21ZR1470600), the Youth Innovation Promotion Association of the Chinese Academy of Sciences (2022264).

## Author contributions

Y. Huang, Lin Z., S.C., and Y.Z. designed and performed the experiments and analyzed the data. R.D. and C.H. contributed to human sample collection and experiment design. Y.S., B.P., X.L., and J.W. provided protocols and suggestions. Y.G., Y. Hu., L.Q., Linyun Z., F.Z., L.Y. provided asthma emergency department visits data. C.Y. and W.G. provided protocols and suggestions. L.M. and Z.L. prepared cell lines and provided reagents. L.P. contributed to UVB data collection. Y. Huang., Y.Z., and B.S. wrote the manuscript. B.S., Y.Z., G.S., L.P., and W.T. supervised the project and revised the manuscript. All authors reviewed the final draft and approved it.

## Competing interests

The authors declare no competing interests.
