## [Peer Review File · Nature Communications]

Solar ultraviolet B radiation promotes α -MSH secretion to attenuate the function of ILC2s via the pituitary–lung axisREVIEWER COMMENTS

Reviewer #1 (Remarks to the Author):

Allergic asthma is regulated by environmental factors such as UVB exposure, but the underlying mechanisms are unclear. ILC2s are important mediators of type 2 inflammation, which is characteristic for atopic disease such as allergic asthma. The authors report a negative correlation between solar UV radiation and asthmatic inflammation in humans. In mice, they show that short term exposure to UVB radiation to the eyes protects from ILC2-mediated lung inflammation in various models. They provide evidence that UVB exposure activates the hypothalamus pituitary-adrenal (HPA) axis, resulting in systemic release of α -melanocyte-stimulating hormone (α -MSH), which negatively regulates ILC2-dependent responses through the Mc5r receptor on ILC2. Mechanistically, α -MSH-dependent Mc5r-signaling downregulates IL7R expression and IL7-signaling in ILC2s. Lastly, the authors provide evidence using PBMCs from asthmatic subjects that ILC2 activation is negatively correlated with plasma levels of α -MSH.

Overall, the idea behind this study is intriguing and most of the data are convincing. However, there are some issues to address:

major points to be addressed:

The HPA axis regulates the response to stress and various biological processes through the production of Glucocorticoids by the adrenal glands upon the release of adrenocorticotrophic hormone (ACTH) by the pituitary gland. ILC2 are well known to express high levels of glucocorticoid receptor (Nr3c1) at much higher levels than Mc5r (see www.immgen.org), thus all the observed effects could also be related to GC effects on ILC2 (Fig 1 to Fig 3). In Fig. 3 it would be useful to measure the serum GC levels. An important issue would be to determine whether the UVB-mediated effect on ILC2-mediated lung inflammation still persists after surgical removal of the adrenal glands.

As mentioned by the authors ILC2 are strongly influenced by sex steroid hormones such as

androgens. Androgens inhibits ILC2 maintenance in barrier tissues in mice, and in human asthma circulating ILC2 have been reported to be lower in males vs females. Sex stratification in Fig. 1 should be mentioned as asthma is more frequent and severe in females vs males. This is already apparent in the Fig. S7 regarding the demographic data of asthmatic patients in Fig. 7 (22 F vs 12 M). The sex of the mice is neither mentioned and this must be corrected.

Does α -MSH levels differentially affect circulating ILC2/IL5 or IL13 according to sex not only in mice but also in humans?

It looks like that Mc5r deletion using IL5-Cre deleter system does not affect ILC2 numbers and phenotype at steady state. However, upon inflammation, it is quite clear that differences exist between WT and Mc5r floxed mice in Figure 4 and Fig S4. Statistical analyses between these two groups are not shown and this has to be done and discussed. This indicates a major effect of endogenous α -MSH at steady state which is present in plasma of normal non UVB sensitized mice (Fig. 3B). What is the mechanism by which Mc5r-signaling regulates ILC2-dependent inflammation in vivo? Is it regulation of proliferation, apoptosis or survival?

Although, it is convincing that α -MSH through Mc5r downregulates IL7R expression and IL7-signaling, is there evidence that this mechanism may affect ILC2 function in a cell-intrinsic manner?

The proposed mechanism of action of α -MSH is unclear. IL7R-signaling is supposed to be important for the survival of ILC2 but not for their effector functions. This mechanistic interpretation should be more thoroughly discussed.

The discussion should be more developed particularly regarding the mechanisms by which α -MSH/Mc5r axis may control ILC2 inflammation and ILC2 effector function versus maintenance/survival in lung tissues.

minor points

There is a problem with the Statistical analyses throughout the paper. In many Figures p values are calculated for group comparison with only two samples or biological replicates. Such analysis cannot be performed on duplicate samples. As it is mentioned in many Figures that the data were representative of two independent experiments, it is unclear why only 2 biological replicates are shown.

Fig. 2F, what does X-axis stand for?

The normalized expression of Mc5r looks very high relative to Hprt. How do the authors calculate this? This does not fit with the low expression level of Mc5r in ILC2 from different tissues using RNA-seq public data set (Ricardo-Gonzalez, (2018). *Nat Immunol* 19(10): 1093-1099).

lane 151: Fig. 2 C,D is mentioned instead of Fig. 3 C,D

Fig. 3F CRH quantification in the hypothalamus. The legend of the left panel is unclear; only n=2 in the right panel. Statistics cannot be informative with n=2.

Fig. 3L How do the author interpret their intracellular staining data for IL5/IL13 after PMA/iono stimulation as PMA/ion bypasses JAK/STAT or Myd88 signaling?

Fig. 4 N-O, The pituitary cell culture system to detect biological activity of endogenous α -MSH is challenging but only 2 biological replicates are shown which is too low to draw any conclusion.

Fig. 6 B and J regarding the quantification of IgE, HDM- or OVA-specific IgE etc.... by ELISA. The way the data are presented is unclear and not described in the Methods.

Fig. 7 legend: Statistical test used are missing as well as the number of PBMCs responding to IL33 stimulation.

lane 259 the first sentence of the discussion a word is missing.

Reviewer #2 (Remarks to the Author):

In this manuscript Huang et al. show the inhibitory effect of UVB light in ILC2s through the α -MSH and MC5R interaction in the context of allergic lung inflammation. The findings are novel and important for the field, the experiments are technically sound and support that ILC2s are inhibited by UVB. However, the implication of these findings in the actual inhibition of the allergic response by solar exposure is not fully demonstrated.

1. The data in Figure 1 showing the association between UVB exposure and allergic symptoms is not novel and overall is weak, as they might be many confounding factors. Thus, this reviewer suggests that this data should be not considered a major finding of this study and included as supporting information.
2. The authors have focused in the pituitary-MSH-ILC2 axis. Have the authors considered an indirect inhibitory effect of UVB exposure on ILC2s? It may be that other cells are affected by UVB which in turn inhibit ILC2s. For instance, UVB activates ILC3s to produce IL-22. Can the authors show the effect of UVB exposure in other cell types within the lung?
3. Figure 2H: It is not clear how the authors conclude that ILC2 present the highest levels of Mc5r in the lung. Sorting criteria should be specified for each population, as ST2+ in the lung include other cells apart from ILC2s.
4. Figure 5: Mc5r- deficient ILC2s are compared to WT ILC2s in stimulating conditions. Where these mice pre-treated with UVB? Otherwise, it is hard to understand the results presented in figure 4, as then super-activation of ILC2s in the Mc5r-KO mice could mask the inhibition by UVB and MSH.
5. Figure 6: In the chronic models, the authors should also show numbers and activation of Th2 cells. Can the authors exclude any role of MSH on T cells by targeting Mc5r exclusively on T cells (Cd4-Cre system).
6. Have the authors tried to give UVB or MSH as a therapeutic approach. It would be very interesting to evaluate the effect of UVB or MSH after the allergen treatment on the activation of ILC2s.

Reviewer #3 (Remarks to the Author):

The immunomodulatory effects of ultraviolet B (UVB) radiation in human diseases have been described and Sun B. and colleagues investigated the cellular molecular mechanisms of how UVB regulates type 2 lung inflammation. In a very elegant study the authors demonstrate here a negative correlation between solar UVB radiation and asthmatic inflammation in humans and mice. UVB exposure to the eyes induced hypothalamus-pituitary activation and α -melanocyte-stimulating hormone (α -MSH) accumulation in the serum to suppress allergic airway inflammation by targeting group 2 innate lymphoid cells (ILC2s) through its receptor MC5R in mice. The α -MSH/MC5R interaction limited ILC2 function. The authors show that one possible mechanism of how α -MSH regulates ILC2 function is through attenuation of JAK/STAT signalling. Consistently, the authors further demonstrate that plasma α -MSH concentration was negatively correlated with the number and function of ILC2s in the peripheral blood mononuclear cells (PBMCs) of asthmatic patients. These findings provide insights into how solar UVB radiation-driven neuroendocrine α -MSH restricts ILC2-mediated lung inflammation and offer a new strategy for controlling allergic diseases.

The study is very comprehensive, elegantly performed and all claims are supported by ample amount of strong and convincing datasets. However, some minor but important questions remain to be addressed and clarified.

1. It would be important if the authors can show protein expression of MC5R by ILC2. Is MC5R expressed by all ILC2 (gut, lung, LN, brain)?

2. Mechanistically the function of MC5R triggering in ILC2 is not fully clear from the data the authors present – some clarification/reinterpretation is required. In Figure 5 and and Figure S5 the authors show that α -MSH negatively regulated CD127 expression in a MC5R-dependent manner. However, the direct impact of the ILC2-activating cytokine IL-33 is not explored. It would be important to demonstrate the role of α -MSH treatment in in vitro experiments after stimulation of naïve lung ILC2s with IL-7 only, IL-2 only, IL-33 only and combinations thereof (IL-7+IL-33, IL-2+IL-33) in the context of α -MSH treatment.

Proliferation, cell viability and type 2 cytokine production should be measured. This set of experiments will yield important mechanistic insights that explain the role of α -MSH

treatment in ILC2 and will corroborate the nice in vivo datasets presented.

3. The authors clearly show the beneficial role of UVB on allergic airway inflammation and type 2 immunopathologies. Most of their interpretation is on MC5R and α -MSH function. However, UVB also induces other ligands that activate aryl hydrocarbon receptor. Specifically, a study by Show and colleagues (2018, Immunity; PMID: 30446384) demonstrates that AhR aryl hydrocarbon receptor signalling inhibits ILC2 functions. Hence, the authors should discuss their findings in a broader context that suggests that UVB through synergistic activation of aryl hydrocarbon receptor and MC5R dampens type 2 immunopathologies.

Point-by-point response to the reviewers' comments

Reviewer #1 (Remarks to the Author):

Allergic asthma is regulated by environmental factors such as UVB exposure, but the underlying mechanisms are unclear. ILC2s are important mediators of type 2 inflammation, which is characteristic for atopic disease such as allergic asthma. The authors report a negative correlation between solar UV radiation and asthmatic inflammation in humans. In mice, they show that short term exposure to UVB radiation to the eyes protects from ILC2-mediated lung inflammation in various models. They provide evidence that UVB exposure activates the hypothalamus pituitary-adrenal (HPA) axis, resulting in systemic release of α -melanocyte-stimulating hormone (α -MSH), which negatively regulates ILC2-dependent responses through the Mc5r receptor on ILC2. Mechanistically, α -MSH-dependent Mc5r-signaling downregulates IL7R expression and IL7-signaling in ILC2s. Lastly, the authors provide evidence using PBMCs from asthmatic subjects that ILC2 activation is negatively correlated with plasma levels of α -MSH.

Overall, the idea behind this study is intriguing and most of the data are convincing. However, there are some issues to address:

Response: Thank you for your appreciation and provided insightful comments for our manuscript.

major points to be addressed:

1. The HPA axis regulates the response to stress and various biological processes through the production of Glucocorticoids by the adrenal glands upon the release of adrenocorticotropic hormone (ACTH) by the pituitary gland. ILC2 are well known to express high levels of glucocorticoid receptor (Nr3c1) at much higher levels than Mc5r (see www.Immgen.org), thus all the observed effects could also be related to GC effects on ILC2 (Fig 1 to Fig 3). In Fig. 3 it would be useful to measure the serum GC levels. An important issue would be to determine whether the UVB-mediated effect on ILC2-mediated lung inflammation still persists after surgical removal of the adrenal glands.

Response:

Thanks for the insightful recommendations. It is a really valuable suggestion to define GC effects on ILC2 in our work. It is well known that Nerve signals as stressors trigger neurons of the paraventricular nucleus (PVN) of the hypothalamus to produce CRH and release it into the pituitary portal circulation (Smith and Vale, 2006; Vale et al., 1981). CRH binds to CRH receptor type 1 (CRH-R1) to induce POMC-derived α -MSH and ACTH into the systemic circulation. Then, ACTH binds to the type II melanocortin receptor (MC2R) in the adrenal cortex to stimulate the synthesis and secretion of glucocorticoids (GCs) (Chrousos, 2009; Miller and Auchus, 2011; Smith and Vale, 2006). GCs (cortisol in humans, corticosterone in rodents) regulate metabolism and immune response. As you mentioned, these observed effects might be related to UVB-CGs on ILC2s.

We performed the experiments to measure the serum corticosterone in the UVB-skin mice (eyes covered) and UVB-eye mice (back skin covered) according to your suggestions (Response Figure 1.A). Our data are consistent with the previous report (Skobowiat and Slominski, 2015), the

exposure of the skin to UVB increased the levels of corticosterone. However, UVB exposure to the eye did not enhance the secretion of corticosterone in mice serum (Response Figure1.B). To compare the function of endogenous α -MSH from the pituitary gland and corticosterone from the adrenal gland under the UVB-eyes irradiation condition, we cultured Mc5r-sufficient and Mc5r-deficient ILC2s in the supernatant of pituitary cells and adrenal gland cells cultures from wild-type (WT) mice with or without UVB irradiation to eyes (Response Figure1.C). The levels of corticosterone did not increase significantly in the culture supernatants of the adrenal gland cells from the UVB-irradiated eyes group (Response Figure1.D). Although we found the stronger inhibitory function of the supernatants from the adrenal gland cells than that from the pituitary cells, the UVB-mediated reduction of type 2 cytokines in ILC2s, which could be abolished after MC5R deletion, was only found in the pituitary cells group, but not in the adrenal gland cells group. (Response Figure1.E).

These data suggest that corticosterone is indeed to inhibit the function of ILC2s, but the UVB-eyes axis involved in our manuscript is more likely to play an inhibitory function in ILC2s through α -MSH/MC5R.

Response Figure 1: Corticosterone concentration and biological activity in UVB-eye axis.

(A) Schematic diagram of the experimental design. (B) The levels of corticosterone in the serum were measured by ELISA 12h after UVB-eye radiation (n= 6 per group). (C) Schematic diagram of the experimental design: ILC2s from $R5^{+/+}$ and $Mc5r^{fl/fl}R5^{+/+}$ female mice cultured with the culture supernatant of pituitary cells or adrenal gland cells from WT mice treated with or without UVB irradiation to the eyes. (D) The levels of corticosterone in the pituitary cell or adrenal gland cell culture supernatant were measured by ELISA (n= 4 per group). (E) The IL-5 and IL-13 protein levels in the supernatants were detected by ELISA (n=2 per sample). The bars and error bars show the means \pm SDs. *, P < 0.05; **, P < 0.01; ***, P < 0.001; ns, not significant; two-tailed Student's t-test and one-way ANOVA with Tukey's multiple comparisons test were used.

2. As mentioned by the authors ILC2 are strongly influenced by sex steroid hormones such as androgens. Androgens inhibits ILC2 maintenance in barrier tissues in mice, and in human asthma

circulating ILC2 have been reported to be lower in males vs females. Sex stratification in Fig. 1 should be mentioned as asthma is more frequent and severe in females vs males. This is already apparent in the Fig. S7 regarding the demographic data of asthmatic patients in Fig. 7 (22 F vs 12 M). The sex of the mice is neither mentioned and this must be corrected. Does 5a-MSH levels differentially affect circulating ILC2/IL5 or IL13 according to sex not only in mice but also in humans?

Response:

Thanks for the insightful recommendations. The sex of the mice had been mentioned in the figure legend in our new manuscript. We had made improvements in our new manuscript in **fig. S1.B**. Sex stratification had been mentioned, and there were no significant differences between females and males in asthma emergency department (ED) visits. Both daily male and daily female asthma ED visits showed a negative correlation with the daily direct solar UVB radiation (fig.S1.B)(Female: $R^2=0.2042$, p -value < 0.0001 ; Male: $R^2=0.1324$, p -value < 0.0001).

Fig. S1. Solar exposure decreased both female and male asthma ED visit rates. (B) Correlation analysis between the average asthma visit rate (daily average) and corresponding UVB radiation from 2015 to 2019 in Shanghai (Left: female; Right: male).

ILC2s are inhibited by sex steroid hormones such as androgens. To explore the influence of sex steroid hormones in the UVB- α -MSH-ILC2s axis, we carried out new experiments, both female mice and male mice were treated with UVB exposure before sensitization by papain administration (fig.S1E). Consistent with previously published reports, the severity of lung inflammation and ILC2s maintenance in male mice were overall lower than in female mice. However, UVB could inhibit lung inflammation in both female and male mice, including eosinophils infiltration (fig.S1, F-G), the levels of the type 2 cytokines IL-5 and IL-13 in the BALF (fig.S1H), and the numbers of ILC2s, IL5⁺ILC2s and IL-13⁺ILC2s in the lungs (fig.S1, I-J).

Fig. S1.: UVB suppressed ILC2-related lung inflammation both in female and male mice. (E) Schematic diagram of the experimental design. WT female and male mice were intratracheally challenged with papain (5 μ g

per mouse) or PBS for 5 consecutive days, and UVB radiation (2.5 kJ/m²) exposure occurred 1 day before sensitization. The mice were sacrificed on day 6. (F-G) Percentage and number of eosinophils in the BALF (Eosinophils, Eos; FVD⁺CD45⁺CD11c^{-lo}SiglecF⁺). (H) The levels of IL-5 and IL-13 in the BALF were measured by ELISA. (I) Statistical analysis of the numbers of ILC2s in the lungs (ILC2s, FVD⁺CD45.2⁺LIN⁻CD90.2⁺CD127⁺ST2⁺). (J) Statistical analysis of the numbers of IL-5⁺IL-13⁺ILC2s, IL-5⁺ILC2s, IL-13⁺ILC2s stimulated with PMA plus ionomycin and BFA for 4 h. The bars and error bars show the means \pm SDs. *, P < 0.05; **, P < 0.01; ***, P < 0.001; ****P < 0.0001; two-tailed Student's t test were used. NS, not significant.

In the clinical samples of figure7, there is no significant difference in the concentration of α -MSH between females and males (Response Figure 2).

Response Figure 2: α -MSH concentration in plasma of female and male asthma patients related to Figure 7. The bars and error bars show the means \pm SDs. *, P < 0.05; **, P < 0.01; ***, P < 0.001; ****P < 0.0001; unpaired t test were used. NS, not significant.

These observations collectively suggested that sex steroid hormones may have no effect on the UVB- α -MSH-ILC2s axis in this work.

3. It looks like that Mc5r deletion using IL5-Cre deleter system does not affect ILC2 numbers and phenotype at steady state. However, upon inflammation, it is quite clear that differences exist between WT and Mc5r floxed mice in Figure 4 and Fig S4. Statistical analyses between these two groups are not shown and this has to be done and discussed. This indicates a major effect of endogenous α -MSH at steady state which is present in plasma of normal non UVB sensitized mice (Fig. 3B). What is the mechanism by which Mc5r-signaling regulates ILC2-dependent inflammation in vivo? Is it regulation of proliferation, apoptosis or survival?

Response:

Thank you for your suggestion, we had made improvements in our revised manuscript including showing the statistical analyses between R5^{+/+} and Mc5r^{fl/fl}R5^{+/+} in Figure 4. Importantly, conditional deletion of the Mc5r gene led to higher severer lung inflammation in mice. In addition to analyzing the differences between R5^{+/+} and Mc5r^{fl/fl}R5^{+/+} mice in Figure 4, we also re-performed the experiments to analyze the difference between the WT and KO groups under inflammatory conditions by using unpaired student's t-test. We found that conditional knockout of MC5R on ILC2 resulted in exacerbated lung inflammation and increased ILC2s response (Response Figure 3).

Response Figure 3: *Mc5r^{fl/fl}R5^{+/+}* mice exhibited exacerbated lung inflammation and increased ILC2s. (A) Schematic diagram of the experimental design. *R5^{+/+}* and *Mc5r^{fl/fl}R5^{+/+}* female mice were intratracheally challenged with papain (4 μ g per mouse) for 5 consecutive days and then sacrificed on day 6. (B) Flow cytometry plots for eosinophils among live CD45⁺ cells in the BALF. (C) Percentage and number of eosinophils in the BALF (n= 6-7 per group). (D) ELISA was performed to measure the levels of IL-5 and IL-13 in the BALF. (E) Representative H&E and PAS staining of lung sections (bars, 200 μ m). (F) Statistical analysis of the percentage and numbers of ILC2s in the lungs. (G) Representative flow diagram showing IL-5 and IL-13 in lung ILC2s stimulated with PMA plus ionomycin and BFA for 4 h. (H) Statistical analysis of the percentages of IL-5⁺ and IL-13⁺ cells among lung ILC2s stimulated with PMA plus ionomycin and BFA for 4 h. The bars and error bars show the means \pm SEMs. *, P < 0.05; **, P < 0.01; ***, P < 0.001; ns, not significant; unpaired Student's t test was used.

These observations may be related to α -MSH autocrine under inflammatory conditions. Previous reports had demonstrated that asthmatic patients have elevated concentrations of α -MSH in BALF compared with healthy individuals (Webering et al., 2019). A similar phenomenon has also been observed in experimental allergic mice (Raap et al., 2003). Therefore, we hypothesized that during the lung inflammation, the systemic production of α -MSH is increased in the human body, consequently, α -MSH can negatively regulate the aggravation of lung inflammation through MC5R on ILC2s, thereby maintaining the homeostasis of the airway inflammation. Therapeutically, administration of α -MSH caused remission of allergic airway inflammation in *R5^{+/+}* mice control littermates, but not in *Mc5r^{fl/fl}R5^{+/+}* mice. Therefore, our work demonstrated that α -MSH suppresses ILC2 responses and allergic airway inflammation through MC5R.

To explore the mechanism by which α -MSH/MC5R axis regulated ILC2s-induced inflammation in vivo, Ki-67 staining was performed to investigate the regulation role of α -MSH/MC5R in ILC2s proliferation. The results showed that α -MSH treatment did not affect the cell proliferation of ILC2s (Response Figure 4). Moreover, in vitro experiments showed that α -MSH treatment did not affect the apoptosis of ILC2s (fig.S8H).

Response Figure 4: The percentages of Ki67⁺ cells among lung ILC2s from R5^{+/+} and Mc5r^{fl/fl}R5^{+/+} female mice with or without daily α -MSH treatment in papain model. The bars and error bars show the means \pm SDs; *, P < 0.05; **, P < 0.01; ***, P < 0.001; ns, not significant; two-tailed Student's t test were used.

Fig. S8. (H) Flow cytometric analysis of PI-Annexin V⁺ ILC2s after 3 days treatment of α -MSH (left). Flow cytometric analysis of PI-Annexin V⁺ ILC2s after 3 days treatment of α -MSH (right). ILC2s cultured with IL-2 (100 U/ml), IL-7 (20 ng/ml), IL-33 (1 ng/ml) respectively, and combinations of IL-2 (100 U/ml) plus IL-33 (1 ng/ml), IL-7 (20 ng/ml) plus IL-33 (1 ng/ml) or IL-2 (100 U/ml) plus IL-7 (20 ng/ml) and IL-33 (1 ng/ml) in the context of α -MSH treatment. The bars and error bars show the means \pm SDs; *, P < 0.05; **, P < 0.01; ***, P < 0.001; ns, not significant; two-tailed Student's t-test were used.

4. Although, it is convincing that α -MSH through Mc5r downregulates IL7R expression and IL7-signaling, is there evidence that this mechanism may affect ILC2 function in a cell-intrinsic manner? The proposed mechanism of action of α -MSH is unclear. IL7R-signaling is supposed to be important for the survival of ILC2 but not for their effector functions. This mechanistic interpretation should be more thoroughly discussed.

Response:

Thanks for the insightful recommendations. As shown in Response Figure 4, conditional deletion of the Mc5r gene led to higher severer lung inflammation in mice. We hypothesized that these observations may be related to α -MSH autocrine under inflammatory conditions. When inflammation occurs, the HP system responds to nerve signals, resulting in systemic production of α -MSH increased, and negatively regulates the aggravation of lung inflammation through MC5R on ILC2s, thereby maintaining the homeostasis of the airway inflammation.

While studies on the function of IL-7/IL-7R signaling in ILC2s are substantial, research on its effector functions are relatively modest. IL-7 was recognized as an important factor in T cell development long before its role in effector function was examined. IL-7 is primarily produced in primary lymphoid tissues, such as bone marrow and thymic stromal cells, where immune cell

development occurs (Hara et al., 2012; Sawa et al., 2009). However, ILC2s respond to multiple cytokines such as IL-33, IL-2, and IL-7, either individually or synergistically to produce effector cytokines such as IL-5 and IL-13 (Halim et al., 2012; Moro et al., 2010; Neill et al., 2010). A recent study showed that IL-7 induces type 2 cytokine response in lung ILC2s and regulates GATA3 and CD25 expression (Sheikh et al., 2022). In our study, we also demonstrated that IL-7 was able to enhance the activation of IL-33 on ILC2 function in vitro (Response Figure 5. A-B). IL-7 and IL-33 have a synergistic role in promoting ILC2 activation. This observation needs further in vivo to define the mechanism of IL7 signaling regulation on ILC2 function. We have discussed this in our new manuscript.

Response Figure 5: IL7-IL-7R signaling amplifies the activation of ILC2 function by IL-33 (A) Flow cytometric analysis of IL-5 and IL-13 in lung ILC2s stimulated with PMA plus ionomycin and BFA for 4 h. ILC2s cultured with IL-33(1 ng/ml) only, IL-7 (20 ng/ml) plus IL-33(1 ng/ml) for 3 days. (B) ELISA was performed to measure IL-5 and IL-13 secretion into the mouse ILC2 culture medium after 3 days. ILC2s cultured with IL-33 IL-33(1 ng/ml) only, IL-7 (20 ng/ml) plus IL-33(1 ng/ml). The bars and error bars show the means \pm SDs. *, $P < 0.05$; **, $P < 0.01$; ***, $P < 0.001$; ns, not significant; unpaired Student's t test was used.

5. The discussion should be more developed particularly regarding the mechanisms by which α -MSH/Mc5r axis may control ILC2 inflammation and ILC2 effector function versus maintenance/survival in lung tissues.

Response:

Thanks for your insightful suggestions. We have reorganized the discussion section and discussed the mechanisms of the α -MSH/Mc5r-IL-7/IL-7R axis in ILC2s (Line356-Lin365).

minor points

There is a problem with the Statistical analyses throughout the paper. In many Figures p values are calculated for group comparison with only two samples or biological replicates. Such analysis cannot be performed on duplicate samples. As it is mentioned in many Figures that the data were representative of two independent experiments, it is unclear why only 2 biological replicates are shown.

Response:

Thank you for this friendly reminder, we had made corrections in our new manuscript. We re-performed some experiments again and increased to more biological replicates, which were shown in Figure.3E, Figure.4O and etc.

Fig. 2F, what does X-axis stand for?

The normalized expression of Mc5r looks very high relative to Hprt. How do the authors calculate this? This does not fit with the low expression level of Mc5r in ILC2 from different tissues using RNA-seq public data set (Ricardo-Gonzalez, (2018). Nat Immunol 19(10): 1093-1099).

Response:

Thanks for pointing this out. Mc5r gene expression was analyzed by qRT-PCR and normalized with HPRT using $2^{-\Delta\Delta Ct}$ method. Compared with other highly expressed genes in ILC2s in other RNA-Seq public databases such as Nmur1 and etc.(Ricardo-Gonzalez et al., 2018), the expression level of Mc5r is not obviously high. However, compared with other melanocortin receptors, Mc1r-4r, the expression of Mc5r on ILC2s is relatively higher. And in the lung, Mc5r is relatively highly expressed in ILC2s compared with other types of immune cells.

lane 151: Fig. 2 C,D is mentioned instead of Fig. 3 C,D

Response:

Thank you for your suggestion, we had made corrections in our new manuscript.

Fig. 3F CRH quantification in the hypothalamus. The legend of the left panel is unclear; only n=2 in the right panel. Statistics cannot be informative with n=2.

Response:

Thank you for this friendly reminder, we had made corrections in our new manuscript. We also modified the legend of Fig.3D-3E to make it clear. In addition, we used a new anti-mouse CRH antibody (abcam, Ab184238) for the western blot analysis of CRH expression changes in the hypothalamus with three biological replicates, and performed the statistical analysis.

Figure 3. (D) Left: representative CRF staining in the hypothalamus under different experimental conditions (CRH: red; DAPI: blue; scale bar,50μm). Right: representative CRH quantification in the hypothalamus. (E) Proteins were comigrated by SDS-PAGE with homogenized hypothalamic tissue and visualized by Western blot analysis with anti-mouse CRH antibodies(above). Statistical analysis of CRH expression was performed using ImageJ (below).

Fig. 3L How do the author interpret their intracellular staining data for IL5/IL13 after PMA/iono stimulation as PMA/ion bypasses JAK/STAT or Myd88 signaling?

Response:

Thanks for pointing this out. PMA plus ionomycin stimulation is a potent, polyclonal and nonspecific method known to induce signaling and subsequent cytokine production. PMA is an analog of diacylglycerol, a key mediator of PKC signaling pathways(Matthews and Cantrell, 2009). Ionomycin stimulates Ca²⁺ release from the ER, activating Ca²⁺-sensitive enzymes and

synergizing with PMA(Chatila et al., 1989). The method of intracellular staining of ILC2s in our study refers to the work of others(Cardoso et al., 2017; Moro et al., 2010), intracellular cytokine staining for IL-13 and IL-5 was performed after stimulation for 4 h with 50 ng/ml PMA and 100 ng/ml ionomycin in the presence of 1 µg/ml Brefeldin A.

Fig. 4 N-O, The pituitary cell culture system to detect biological activity of endogenous α -MSH is challenging but only 2 biological replicates are shown which is too low to draw any conclusion.

Response:

Thanks for pointing this out. We performed new experiments with four biological replicates. Consistent with the observations drawn in the previous manuscript, the levels of α -MSH were higher in the culture supernatants of mouse pituitary cells from mice with UVB-irradiated eyes (Fig. 4O, left), and the ability of supernatants of pituitary cells from mice with UVB- irradiated eyes to inhibit ILC2 secretion of type 2 cytokines was more robust than that of those from sham-irradiated mice, but the inhibition was not observed in *Mc5r*-deficient ILC2s group. (Fig. 4O, right).

Figure 4. (O) The levels of α -MSH in the pituitary cell culture supernatant were measured by ELISA (n= 4 per group); the IL-5 and IL-13 protein levels in the supernatants were detected by ELISA (n=4 per sample). The bars and error bars show the means \pm SDs. *, P < 0.05; **, P < 0.01; ***, P < 0.001; ns, not significant; unpaired Student's t-test two-tailed Student's t-test were used. The data are representative of two or more independent experiments.

Fig. 6 B and J regarding the quantification of IgE, HDM- or OVA-specific igE etc.... by ELISA. The way the data are presented is unclear and not described in the Methods.

Response:

Thank you for pointing this out, we had made improvements in our new manuscript to describe the ELISA analysis of IgE, HDM- or OVA-specific IgE in the Method section.

Fig. 7 legend: Statistical test used are missing as well as the number of PBMCs responding to IL33 stimulation.

Response:

Thank you for this friendly reminder, we had made corrections in our new manuscript.

lane 259 the first sentence of the discussion a word is missing.

Response:

Thank you for your suggestion, we had made improvements in our new manuscript.

Reviewer #2 (Remarks to the Author):

In this manuscript Huang et al. show the inhibitory effect of UVB light in ILC2s through the α -MSH and MC5R interaction in the context of allergic lung inflammation. The findings are novel and important for the field, the experiments are technically sound and support that ILC2s are inhibited by UVB. However, the implication of these findings in the actual inhibition of the allergic response by solar exposure is not fully demonstrated.

1. The data in Figure 1 showing the association between UVB exposure and allergic symptoms is not novel and overall is weak, as they might be many confounding factors. Thus, this reviewer suggests that this data should be not considered a major finding of this study and included as supporting information.

Response:

Thank you for your suggestion, we had made improvements in our new manuscript.

2. The authors have focused in the pituitary-MSH-ILC2 axis. Have the authors considered an indirect inhibitory effect of UVB exposure on ILC2s? It may be that other cells are affected by UVB which in turn inhibit ILC2s. For instance, UVB activates ILC3s to produce IL-22. Can the authors show the effect of UVB exposure in other cell types within the lung?

Response:

Thanks for your insightful suggestions. We detected the differences among other cell types in the lung, such as neutrophils, macrophages (fig.S2B), T cells, B cells, natural killer cells (fig.S2C), and dendritic cells (fig.S2D) between nonirradiated and UVB-irradiated mice with papain stimulation. These results showed that eosinophils and ILC2s were reduced after UVB radiation (fig.S2E), and the other cell types remain unaffected (fig.S2, F-H).

Fig. S2. Myeloid and lymphoid cell distributions in the lungs of mice after exposure to UVB.

(A) Schematic diagram of the experimental design. WT female mice were intratracheally challenged with papain (5 µg per mouse) for 5 consecutive days, and UVB radiation (2.5 kJ/m²) exposure occurred 1 day before sensitization. The mice were sacrificed on day 6. (B) Gating strategy for lung neutrophils (Neu: FVD⁻CD45⁺Ly6G⁺), eosinophils (Eos: FVD⁻CD45⁺Ly6G⁻Siglec-F⁺CD11c⁻), macrophages (MAC: FVD⁻CD45⁺Ly6G⁻Siglec-F⁺CD11c⁺). (C) Gating strategy for lung natural killer (NK) cells (FVD⁻CD45⁺CD3⁻NKp46⁺), T cells (FVD⁻CD45⁺CD3⁺), and B cells (FVD⁻CD45⁺CD3⁻NKp46⁻CD19⁺). (D) Gating strategy for lung CD103⁺ DCs (FVD⁻CD45⁺CD11b⁻CD103⁺CD11c⁺MHC class II⁺) and CD11b⁺ DCs (FVD⁻CD45⁺CD11b⁺CD103⁻CD11c⁺MHC class II⁺). (E) The number of eosinophils and ILC2s in lung from papain challenged mice with or without UVB treatment. (F) Number of macrophages and neutrophils in lung from papain challenged mice with or without UVB treatment. (G) Number of NKs, T cells, B cells in lung from papain challenged mice with or without UVB treatment. (H) Number of CD103⁺ DCs and CD11b⁺ DCs in lung from papain challenged mice with or without UVB treatment. The bars and error bars show the means ± SDs. *, P < 0.05; **, P < 0.01; ***, P < 0.001; ****P < 0.0001; unpaired t test were used. NS, not significant. Data are representative of two or more independent experiments.

Chronic UVB could induce skin RORγt⁺ ILC (uvILC3) that produces interleukin-22 locally and is associated with mutant skin cell growth (Lewis et al., 2015; Lewis et al., 2021). ILC3s are also implicated in gastrointestinal immune responses (Zeng et al., 2019). We detected the number of ILC2s and ILC3s in mesenteric lymph nodes and colon propria between nonirradiated and UVB-irradiated mice and found both of them remained unaffected (Response Figure 6).

Response Figure 6 : ILC2 and ILC3 distributions in the mesenteric lymph nodes and colon of mice in steady state after exposure to UVB.

(A) Gating strategy for ILC2s (FVD⁻CD45⁺Lin⁻CD127⁺KLRG1⁺) and ILC3s (FVD⁻CD45⁺Lin⁻CD127⁺RORγt⁺) in the mesenteric lymph nodes (mLN). (B) The number of ILC2s and ILC3s in mLN from naïve mice with or without UVB treatment. (C) The number of ILC2s and ILC3s in colon from naïve mice with or without UVB treatment. The bars and error bars show the means ± SDs. *, P < 0.05; **, P < 0.01; ***, P < 0.001; ****P < 0.0001; unpaired t test were used. NS, not significant.

3. Figure 2H: It is not clear how the authors conclude that ILC2 present the highest levels of Mc5r in the lung. Sorting criteria should be specified for each population, as ST2⁺ in the lung include other cells apart from ILC2s.

Response:

Thank you for your suggestion, we had made improvements in our new manuscript. Sorting criteria had been for each population in the legend of Figure 2H.

4. Figure 5: Mc5r- deficient ILC2s are compared to WT ILC2s in stimulating conditions. Where these mice pre-treated with UVB? Otherwise, it is hard to understand the results presented in figure 4, as then super-activation of ILC2s in the Mc5r-KO mice could mask the inhibition by UVB and

MSH.

Response:

Thank you for the friendly reminder. We agree that adding the RNA-seq of ILC2s from mice pre-treated with UVB would be useful to understand the details. However, we believe the present results can still support the conclusion of our work. As shown in Response Figure 3, conditional deletion of the *Mc5r* gene led to higher severer lung inflammation in mice. Those observations may be related to α -MSH autocrine under inflammatory conditions. Previous reports had demonstrated that asthmatic patients have elevated concentrations of α -MSH in BALF compared with healthy individuals(Webering et al., 2019). A similar phenomenon has also been observed in experimental allergic mice(Raap et al., 2003). Therefore, we hypothesized that during lung inflammation, a systemic production of α -MSH is increased, and it negatively regulates the aggravation of lung inflammation through MC5R on ILC2s, thereby maintaining the homeostasis of the airway inflammation. Similarly, in Figure 3B, we can also detect the presence of α -MSH in the resting state (UVB untreated), further suggesting the possibility that the physiological levels of α -MSH maintain inflammatory homeostasis. Therapeutically, UVB treatment and administration of α -MSH both caused remission of allergic airway inflammation in *Mc5r^{+/+}R5/+* mice control littermates, but not in *Mc5r^{fl/fl}R5/+* mice. Therefore, we speculate that both the increased α -MSH caused by inflammation and the α -MSH stimulated by UVB may exert inhibitory functions through MC5R on the surface of ILC2s.

Response Figure 3: *Mc5r^{fl/fl}R5/+* mice exhibited exacerbated lung inflammation and increased ILC2s.

(A) Schematic diagram of the experimental design. *Mc5r^{+/+}R5/+* and *Mc5r^{fl/fl}R5/+* female mice were intratracheally challenged with papain (4 μ g per mouse) for 5 consecutive days and then sacrificed on day 6. (B) Flow cytometry plots for eosinophils among live CD45⁺ cells in the BALF. (C) Percentage and number of eosinophils in the BALF (n= 6-7 per group). (D) ELISA was performed to measure the levels of IL-5 and IL-13 in the BALF. (E) Representative H&E and PAS staining of lung sections (bars, 200 μ m). (F) Statistical analysis of the percentage and numbers of ILC2s in the lungs. (G) Representative flow diagram showing IL-5 and IL-13 in lung ILC2s stimulated with PMA plus ionomycin and BFA for 4 h. (H) Statistical analysis of the percentages of IL-5⁺ and IL-13⁺ cells among lung ILC2s stimulated with PMA plus ionomycin and BFA for 4 h. The bars and error bars show

the means \pm SEMs. *, $P < 0.05$; **, $P < 0.01$; ***, $P < 0.001$; ns, not significant; unpaired Student's t test was used.

Figure 3. (B) The levels of α -MSH in the serum were measured by ELISA (n= 6-7 per group).

5. Figure 6: In the chronic models, the authors should also show numbers and activation of Th2 cells. Can the authors exclude any role of MSH on T cells by targeting Mc5r exclusively on T cells (Cd4-Cre system).

Response:

Thank you for your suggestion, we had made improvements in our new manuscript including showing the number of $ST2^+CD4^+$ T cells and functional $ST2^+CD4^+$ T cells (Figure.6I-J) in HDM-induced chronic lung inflammation. The numbers and activation of Th2 cells were not affected by α -MSH in the HDM model. However, the expression of MC5R was low in $CD4^+$ T cells compared to ILC2s (Figure.2H), so we hypothesized that the effect of α -MSH on T cells is limited.

Figure 6. (I) The number of $ST2^+CD4^+$ T cells in the lungs. (J) The number of $IL-5^+IL-13^+$ cells among lung $ST2^+CD4^+$ T cells after stimulation with PMA, ionomycin, and BFA for 4 h. The bars and error bars show the means \pm SDs. *, $P < 0.05$; **, $P < 0.01$; ***, $P < 0.001$; ns, not significant; unpaired Student's t test.

Figure 2. (H) Relative expression of *Mc5r* in the indicated populations of sorted immune cells from the lungs (B

cells:FVD⁻CD3⁻CD19⁺;CD4⁺cells: FVD⁻CD3⁺CD4⁺;CD45⁺cells: FVD⁻CD45⁺; CD8⁺cells: FVD⁻CD3⁻CD8⁺;DCs: FVD⁻CD45⁺CD103⁺MHC-II⁺CD11c⁺; Eosinophils:FVD⁻CD45⁺CD11c^{-/lo}SiglecF⁺; Alveolar macrophage:AMs, FVD⁻CD45⁺Ly6G⁻CD11c⁺SiglecF⁺; Interstitial macrophages(IM: CD45⁺Ly6G⁻SiglecF⁻CD11c⁺CD11b⁺F4/80⁺;Neutrophils: FVD⁻CD45⁺Ly6G⁺CD11b⁺;ILC2s: FVD⁻CD45.2⁺LIN⁻CD90.2⁺CD127⁺ST2⁺).

6. Have the authors tried to give UVB or MSH as a therapeutic approach. It would be very interesting to evaluate the effect of UVB or MSH after the allergen treatment on the activation of ILC2s.

Response:

Thanks for the valuable suggestions to improve our work. It would be important to explore whether α -MSH can intervene in ILC2-mediated type 2 immune response at a later stage. We treated WT female mice with papain for 5 days and then administered α -MSH intratracheally on day 6 to day 8. Lung inflammation and ILC2 function were measured on day 10(Fig. 7F). Interestingly, the percentage and number of eosinophils were reduced in BALF from α -MSH-treated mice (Fig. 7G-H). α -MSH treatment also ameliorated inflammatory cell infiltration and mucus hypersecretion (Fig. 7I). Moreover, α -MSH-treated mice exhibited decreased ILC2 numbers in the lung (Fig. 7J). Intracellular staining of type 2 cytokines revealed a significant reduction in the percentage of IL-5- and IL-13-producing ILC2s (Fig. 7K-L). These observations indicate that α -MSH has the potential in the treatment of ILC2-induced type 2 inflammatory diseases.

Fig.7. α -MSH showed an effective intervention of ILC2 mediated type 2 immune response.

(F) Schematic diagram of the experimental design. The WT female mice were intratracheally challenged with papain for 5 consecutive days and intratracheally treated with PBS or α -MSH (4 mg/kg) for two consecutive days and sacrificed on day 10. (G) Representative flow diagram showing the percentage of eosinophils in the BALF. (H) Percentage and number of eosinophils in the BALF. (I) Statistical analysis of the numbers of ILC2s in the

lungs. (I) Representative HE and PAS staining of lung sections (bars, 200 μ m). (K) Flow cytometric analysis of IL-5 and IL-13 in lung ILC2s stimulated with PMA plus ionomycin and BFA for 4 h. (L) Statistical analysis of the percentages of IL-5⁺ and IL-13⁺ cells among lung ILC2s. The bars and error bars show the means \pm SDs. Unpaired t test and Pearson's correlation coefficient analysis were used. *, P < 0.05; **, P < 0.01; ***, P < 0.001; ****P < 0.0001 NS, not significant.

Reviewer #3 (Remarks to the Author):

The immunomodulatory effects of ultraviolet B (UVB) radiation in human diseases have been described and Sun B. and colleagues investigated the cellular molecular mechanisms of how UVB regulates type 2 lung inflammation. In a very elegant study the authors demonstrate here a negative correlation between solar UVB radiation and asthmatic inflammation in humans and mice. UVB exposure to the eyes induced hypothalamus-pituitary activation and α -melanocyte-stimulating hormone (α -MSH) accumulation in the serum to suppress allergic airway inflammation by targeting group 2 innate lymphoid cells (ILC2s) through its receptor MC5R in mice. The α -MSH/MC5R interaction limited ILC2 function. The authors show that one possible mechanism of how α -MSH regulates ILC2 function is through attenuation of JAK/STAT signalling. Consistently, the authors further demonstrate that plasma α -MSH concentration was negatively correlated with the number and function of ILC2s in the peripheral blood mononuclear cells (PBMCs) of asthmatic patients. These findings provide insights into how solar UVB radiation-driven neuroendocrine α -MSH restricts ILC2-mediated lung inflammation and offer a new strategy for controlling allergic diseases.

The study is very comprehensive, elegantly performed and all claims are supported by ample amount of strong and convincing datasets. However, some minor but important questions remain to be addressed and clarified.

1. It would be important if the authors can show protein expression of MC5R by ILC2. Is MC5R expressed by all ILC2 (gut, lung, LN, brain)?

Response:

We sincerely appreciate the valuable suggestions. To show the protein expression of MC5R by ILC2, we carried out the experiments according to your suggestion but did not get the desired results because of the inappropriate antibody we selected. MC5R is a cell surface receptor that belongs to the class of G-protein coupled receptors (GPCRs). It is difficult for us to choose a specific antibody to detect the protein level of MC5R by ILC2s.

We sorted murine ILC2s from different tissues such as the lung, brain (including meninges), and mesenteric lymph nodes (mLN) to detect the transcriptional level of *Mc5r*. We also sorted KLRG1⁺ ILC from mLN as a control. The results showed that MC5R was expressed in ILC2s from lung, brain, and mesenteric lymph nodes, but only weakly expressed in KLRG1⁺ ILC. These observations suggested that the role of the MC5R/ILC2 axis in other ILC2s-mediated diseases needs to be further characterized.

Response Figure 7: qRT-PCR analysis of Mc5r expression in ILC2s from different tissue.

Lung ILC2s and brain ILC2s: FVD-CD45.2⁺LIN⁻CD90.2⁺CD127⁺ST2⁺; mLN ILC2s: FVD-CD45.2⁺LIN⁻CD90.2⁺CD127⁺KLRG1⁺; none-ILC2: FVD-CD45.2⁺LIN⁻CD90.2⁺CD127⁺KLRG1⁻

2. Mechanistically the function of MC5R triggering in ILC2 is not fully clear from the data the authors present – some clarification/reinterpretation is required. In Figure 5 and and Figure S5 the authors show that α -MSH negatively regulated CD127 expression in a MC5R-dependent manner. However, the direct impact of the ILC2-activating cytokine IL-33 is not explored. It would be important to demonstrate the role of α -MSH treatment in in vitro experiments after stimulation of naïve lung ILC2s with IL-7 only, IL-2 only, IL-33 only and combinations thereof (IL-7+IL-33, IL-2+IL-33) in the context of α -MSH treatment. Proliferation, cell viability and type 2 cytokine production should be measured. This set of experiments will yield important mechanistic insights that explain the role of α -MSH treatment in ILC2 and will corroborate the nice in vivo datasets presented.

Response:

Thanks for the valuable suggestions to improve our work. It would be important to explore the role of α -MSH in the cytokine activation pathway of ILC2s. According to your suggestions, we cultured ILC2s under different cytokine conditions in the context of α -MSH treatment. In the culture condition of IL-2, IL-7 or IL-33 respectively, the treatment of α -MSH did not significantly inhibit the function of ILC2s to release cytokines (Fig.5I). However, the inhibitory function of α -MSH was effective under the condition of IL-7 plus IL-33 but not IL-2 plus IL-33 (Fig.5J). And in the presence of all three cytokines, the inhibitory function of α -MSH on ILC2 was amplified (Fig.5J). Under the corresponding conditions, the apoptosis and proliferation of ILC2 were not significantly affected by α -MSH (fig.S8J). In response to this phenomenon, we suspected that IL-33 could up-regulate the expression of CD127 (fig.S8K), enhance the signal of IL-7/IL-7R, and thus amplify the inhibitory function of α -MSH. And the protein expression level of p65 (phosphorylated forms) in NF- κ B pathways was also downregulated by α -MSH treatment in IL-7 plus IL-33 condition (Fig.5K). With these findings, α -MSH serves as a potent inhibitory signal for JAK/STAT and NF- κ B pathways to regulate type 2 cytokines in lung ILC2s.

Figure 5. (I) ELISA was performed to measure IL-5 and IL-13 secretion into the mouse ILC2 culture medium after 3 days. ILC2s cultured with IL-2 (100 U/ml), IL-7(20 ng/ml) or IL-33(1 ng/ml) respectively in the context of α -MSH treatment. (J) ELISA was performed to measure IL-5 and IL-13 secretion into the mouse ILC2 culture medium after 3 days. ILC2s cultured in combinations of IL-2 (100 U/ml) plus IL-33(1 ng/ml), IL-7(20 ng/ml) plus IL-33(1 ng/ml) or IL-2 (100 U/ml) plus IL-7(20 ng/ml) and IL-33(1 ng/ml) in the context of α -MSH treatment. (K) Sorted ILC2s were cultured in the presence or absence of α -MSH for 3 days, then cultured in the absence of cytokines for 24 h and re-stimulated with IL-7, IL-33, IL-7 plus IL-33 for 0, 10 and 60 min for Western blot analysis with the indicated antibodies(up). Statistical analysis of the levels of p65 and STAT5 phosphorylation (p-65 and p-STAT5, respectively) was performed using ImageJ(down). The bars and error bars show the means \pm SDs; *, $P < 0.05$; **, $P < 0.01$; ***, $P < 0.001$; ns, not significant; two-way ANOVA with Tukey's multiple comparisons test and two-tailed Student's t test were used.

Fig. S8. (J) Flow cytometric analysis of PI-Annexin V⁺ILC2s after 3 days treatment of α -MSH (up). Flow cytometric analysis of PI-Annexin V⁺ILC2s after 3 days treatment of α -MSH (down). ILC2s cultured with IL-2 (100 U/ml), IL-7(20 ng/ml), IL-33(1 ng/ml) respectively, and combinations of IL-2 (100 U/ml) plus IL-33(1 ng/ml), IL-7(20 ng/ml) plus IL-33(1 ng/ml) or IL-2 (100 U/ml) plus IL-7(20 ng/ml) and IL-33(1 ng/ml) in the context of α -MSH treatment. (K) Flow cytometric analysis of CD127 in ILC2s under the different described experimental conditions. The bars and error bars show the means \pm SDs; *, $P < 0.05$; **, $P < 0.01$; ***, $P < 0.001$; ns, not significant; two-tailed Student's t-test were used.

3. The authors clearly show the beneficial role of UVB on allergic airway inflammation and type 2 immunopathologies. Most of their interpretation is on MC5R and α -MSH function. However, UVB also induces other ligands that activate aryl hydrocarbon receptor. Specifically, a study by Show and colleagues (2018, *Immunity*; PMID: 30446384) demonstrates that AhR aryl hydrocarbon receptor signalling inhibits ILC2 functions. Hence, the authors should discuss their findings in a broader context that suggests that UVB through synergistic activation of aryl hydrocarbon receptor and MC5R dampens type 2 immunopathologies.

Response:

Thanks for this insightful recommendation. It is a valuable suggestion to improve the quality of our manuscript. According to your suggestion, we have reorganized the discussion section and discussed the possibility that UVB may inhibit ILC2-mediated lung inflammation in a complex process orchestrating AhR activation and α -MSH/MC5R signaling (Line321-327).

References

- Cardoso, V., Chesné, J., Ribeiro, H., García-Cassani, B., Carvalho, T., Bouchery, T., Shah, K., Barbosa-Morais, N.L., Harris, N., and Veiga-Fernandes, H. (2017). Neuronal regulation of type 2 innate lymphoid cells via neuromedin U. *Nature* 549, 277-281.
- Chatila, T., Silverman, L., Miller, R., and Geha, R. (1989). Mechanisms of T cell activation by the calcium ionophore ionomycin. *J Immunol* 143, 1283-1289.
- Chrousos, G.P. (2009). Stress and disorders of the stress system. *Nat Rev Endocrinol* 5, 374-381.
- Halim, T.Y., Krauss, R.H., Sun, A.C., and Takei, F. (2012). Lung natural helper cells are a critical source of Th2 cell-type cytokines in protease allergen-induced airway inflammation. *Immunity* 36, 451-463.
- Hara, T., Shitara, S., Imai, K., Miyachi, H., Kitano, S., Yao, H., Tani-ichi, S., and Ikuta, K. (2012). Identification of IL-7-producing cells in primary and secondary lymphoid organs using IL-7-GFP knock-in mice. *J Immunol* 189, 1577-1584.
- Lewis, J.M., Bürgler, C.D., Freudzon, M., Golubets, K., Gibson, J.F., Filler, R.B., and Girardi, M. (2015). Langerhans Cells Facilitate UVB-Induced Epidermal Carcinogenesis. *J Invest Dermatol* 135, 2824-2833.
- Lewis, J.M., Monico, P.F., Mirza, F.N., Xu, S., Yumeen, S., Turban, J.L., Galan, A., and Girardi, M. (2021). Chronic UV radiation-induced ROR γ t⁺ IL-22-producing lymphoid cells are associated with mutant KC clonal expansion. *Proc Natl Acad Sci U S A* 118.
- Matthews, S.A., and Cantrell, D.A. (2009). New insights into the regulation and function of serine/threonine kinases in T lymphocytes. *Immunol Rev* 228, 241-252.
- Miller, W.L., and Auchus, R.J. (2011). The molecular biology, biochemistry, and physiology of human steroidogenesis and its disorders. *Endocr Rev* 32, 81-151.
- Moro, K., Yamada, T., Tanabe, M., Takeuchi, T., Ikawa, T., Kawamoto, H., Furusawa, J., Ohtani, M., Fujii, H., and Koyasu, S. (2010). Innate production of T(H)2 cytokines by adipose tissue-associated c-Kit(+)Sca-1(+) lymphoid cells. *Nature* 463, 540-544.
- Neill, D.R., Wong, S.H., Bellosi, A., Flynn, R.J., Daly, M., Langford, T.K., Bucks, C., Kane, C.M., Fallon, P.G., Pannell, R., et al. (2010). Nuocytes represent a new innate effector leukocyte that mediates type-2 immunity. *Nature* 464, 1367-1370.
- Raap, U., Brzoska, T., Sohl, S., Páth, G., Emmel, J., Herz, U., Braun, A., Luger, T., and Renz, H. (2003). Alpha-melanocyte-stimulating hormone inhibits allergic airway inflammation. *J Immunol* 171, 353-359.
- Ricardo-Gonzalez, R.R., Van Dyken, S.J., Schneider, C., Lee, J., Nussbaum, J.C., Liang, H.E., Vaka, D., Eckalbar, W.L., Molofsky, A.B., Erle, D.J., et al. (2018). Tissue signals imprint ILC2 identity with

anticipatory function. *Nat Immunol* *19*, 1093-1099.

Sawa, Y., Arima, Y., Ogura, H., Kitabayashi, C., Jiang, J.J., Fukushima, T., Kamimura, D., Hirano, T., and Murakami, M. (2009). Hepatic interleukin-7 expression regulates T cell responses. *Immunity* *30*, 447-457.

Sheikh, A., Lu, J., Melese, E., Seo, J.H., and Abraham, N. (2022). IL-7 induces type 2 cytokine response in lung ILC2s and regulates GATA3 and CD25 expression. *J Leukoc Biol* *112*, 1105-1113.

Skobowiat, C., and Slominski, A.T. (2015). UVB Activates Hypothalamic-Pituitary-Adrenal Axis in C57BL/6 Mice. *J Invest Dermatol* *135*, 1638-1648.

Smith, S.M., and Vale, W.W. (2006). The role of the hypothalamic-pituitary-adrenal axis in neuroendocrine responses to stress. *Dialogues Clin Neurosci* *8*, 383-395.

Vale, W., Spiess, J., Rivier, C., and Rivier, J. (1981). Characterization of a 41-residue ovine hypothalamic peptide that stimulates secretion of corticotropin and beta-endorphin. *Science* *213*, 1394-1397.

Webering, S., Lunding, L.P., Vock, C., Schröder, A., Gaede, K.I., Herzmann, C., Fehrenbach, H., and Wegmann, M. (2019). The alpha-melanocyte-stimulating hormone acts as a local immune homeostasis factor in experimental allergic asthma. *Clin Exp Allergy* *49*, 1026-1039.

Zeng, B., Shi, S., Ashworth, G., Dong, C., Liu, J., and Xing, F. (2019). ILC3 function as a double-edged sword in inflammatory bowel diseases. *Cell Death Dis* *10*, 315.

REVIEWERS' COMMENTS

Reviewer #1 (Remarks to the Author):

The authors have responded to most of my concerns and performed additional experiments to clarify some points. However, they are still some concerns that need to be addressed.

Although the authors do not performed adrenalectomy to exclude a role of GC, they measure serum GC content and show that UVB exposure to the eyes did not show enhanced GC secretion. Using a complex in vitro culture model, they show that the α -MSH/MC5R axis contributes to ILC2 inhibition when co-cultured with pituitary cells from UVB-eyes mice. These data shown as response Fig. 1 should be added to ms as a supplementary figure. In the Fig legend what does n=2 means in panel (E) as more than 2 samples are shown?

The effect of biological sex has been now taken into consideration in the revised manuscript. Sex of the mice is now specified in the manuscript and a new Supplementary Fig. 3 has been added comparing male and female mice.

The data presented as "response Fig. 2" (α -MSH concentration in the serum of male or female asthmatic patients) should be added to the supplementary Fig 11.

The observation that α -MSH suppresses ILC2 responses and allergic airway inflammation through MC5R without UVB irradiation (response Fig. 3) is an important observation and should be shown in the manuscript as supplementary figure related to Figure 4.

Regarding the mechanism the authors have addressed the points I raised. This is not due to inhibition of cell proliferation nor apoptosis. They show a synergistic action of IL7/IL33 axis to ILC2 activation which sustain their hypothesis that that inhibition of IL7ra expression by α -MSH/MC5R axis is a convincing mechanism to explain the protective effect of α -MSH on ILC2-mediated lung inflammation as shown in Figure 5.

Figure 5 statistics still need to be corrected in panel (K). I understand that western blot analysis of phospho-p65 and phosphor-STAT5 from lung ILC2 activated by various

combination of cytokines (IL7, IL33 or both) are very challenging experiments, but SD and statistics cannot be performed on n=2 !!!.

So, I suggest to the authors to simply remove the SD and the p values, and to show the histogram with the two independent experiments. This is convincing enough.

There is still a problem with the discussion line 358-367 & 347-357. These two part are not easy to follow and must be re-written.

Reviewer #2 (Remarks to the Author):

I really appreciate all the efforts made by the authors to respond to our questions. I am satisfied with the new experiments done and the revised text. I believe it is a piece of work that will contribute positively to our field.

Reviewer #3 (Remarks to the Author):

The authors have addressed all comments raised by this reviewer sufficiently well and also addressed several key points raised by other evaluators.

REVIEWERS' COMMENTS

Reviewer #1 (Remarks to the Author):

The authors have responded to most of my concerns and performed additional experiments to clarify some points. However, they are still some concerns that need to be addressed.

Although the authors do not performed adrenalectomy to exclude a role of GC, they measure serum GC content and show that UVB exposure to the eyes did not show enhanced GC secretion. Using a complex in vitro culture model, they show that the α -MSH/MC5R axis contributes to ILC2 inhibition when co-cultured with pituitary cells from UVB-eyes mice. These data shown as response Fig. 1 should be added to ms as a supplementary figure. In the Fig legend what does n=2 means in panel (E) as more than 2 samples are shown?

Response: Thank you for your appreciation and provided insightful comments for our manuscript. We had added response Fig1 to MS as supplementary figure 9. Thank you for this friendly reminder, we had made corrections in the Fig legend in our new version.

The effect of biological sex has been now taken into consideration in the revised manuscript. Sex of the mice is now specified in the manuscript and a new Supplementary Fig. 3 has been added comparing male and female mice.

Response: Thank you for your appreciation.

The data presented as “response Fig. 2” (α -MSH concentration in the serum of male or female asthmatic patients) should be added to the supplementary Fig 11.

Response: Thank you for your suggestion, we had added response Fig2 to MS as supplementary figure 13b.

The observation that α -MSH suppresses ILC2 responses and allergic airway inflammation through MC5R without UVB irradiation (response Fig. 3) is an important observation and should be shown in the manuscript as supplementary figure related to Figure 4.

Response: Thank you for your suggestion, we had added response Fig3 to MS as supplementary figure 8.

Regarding the mechanism the authors have addressed the points I raised. This is not due to inhibition of cell proliferation nor apoptosis. They show a synergistic action of IL7/IL33 axis to ILC2 activation which sustain their hypothesis that that inhibition of Il7ra expression by α -MSH/MC5R axis is a convincing mechanism to explain the protective effect of α -MSH on ILC2-mediated lung inflammation as shown in Figure 5.

Response: Thank you for your appreciation.

Figure 5 statistics still need to be corrected in panel (K). I understand that western blot analysis of phospho-p65 and phosphor-STAT5 from lung ILC2 activated by various combination of cytokines (IL7, IL33 or both) are very challenging experiments, but SD and statistics cannot be performed on n=2 !!!.

So, I suggest to the authors to simply remove the SD and the p values, and to show the histogram with the two independent experiments. This is convincing enough.

Response: Thank you for this friendly reminder, we had made corrections in in our new version according to your suggestions.

There is still a problem with the discussion lane 358-367 & 347-357. These two spart are not easy to follow and must be re-written.

Response: Thanks for this insightful recommendation. It is a valuable suggestion to improve the quality of our manuscript. According to your suggestion, we have modified the discussion section.

Reviewer #2 (Remarks to the Author):

I really appreciate all the efforts made by the authors to respond to our questions. I am satisfied with the new experiments done and the revised text. I believe it is a piece of work that will contribute positively to our field.

Response: Thank you for your appreciation.

Reviewer #3 (Remarks to the Author):

The authors have addressed all comments raised by this reviewer sufficiently well and also addressed several key points raised by other evaluators.

Response: Thank you for your appreciation.